# LEARNING LINEAR DYNAMICAL SYSTEMS WITH SPARSE SYSTEM MATRICES

## ABSTRACT

Due to the tractable analysis and control, linear dynamical systems (LDSs) provide a fundamental mathematical tool for time-series data modeling in various disciplines. Particularly, many LDSs have sparse system matrices because interactions among variables are limited or only a few significant relationships exist. However, available learning algorithms for LDSs lack the ability to learn system matrices with the sparsity constraint. To address this issue, we impose sparsity-promoting priors on system matrices and explore the expectation–maximization (EM) algorithm to give a maximum a posteriori (MAP) estimate of both hidden states and system matrices from noisy observations. In addition, we find that many learning algorithms based on the gradient descent method use an inappropriate derivative rule, because they neglect the inherent symmetry of noise covariance matrices. Here, we consider the derivative rule of structured matrices during the optimization process to guarantee their symmetry. Experimental results on simulation and real-world problems illustrate that the proposed algorithm significantly improves learning accuracy over classical ones.

## 1 INTRODUCTION

Linear dynamical systems (LDSs) are fundamental mathematical models for analyzing time-series data with application in robotics (Mamakoukas et al., 2019; 2020), systems biology (Jin et al., 2020b; Pillonetto & Ljung, 2023), and natural language processing (Smith et al., 1999; Belanger & Kakade, 2015). Basically, LDSs consider a linear transformation between finite-dimensional hidden states disturbed by input signals and noise to capture the time evolution of systems (Hazan et al., 2017). In addition, LDSs are also widely used to approximate complex nonlinear systems in industrial processes given their relative simplicity (Yuan et al., 2017; Lusch et al., 2018). Due to a complete rigorous theory available on LDSs, learning LDSs from noisy observations can enable us to make tractable analysis and control of systems (Chen & Poor, 2022; Bakshi et al., 2023).

In this paper, we focus on learning LDSs with sparse system matrices for two important reasons. First, the learned LDSs should include the minimally required parameters to explain time-series data following the *Occam's razor* principle. Additionally, many real-world systems have sparse topology because each state or measurement variable only depends on a few other state variables and inputs (Efroni et al., 2022). For example, a gene only regulates the expression of a limited number of other genes in gene regulatory networks (He et al., 2024). In industry, communication systems usually have sparse topology to reduce energy consumption (Jin et al., 2020a;b). However, available learning algorithms lack the ability to learn LDSs with the sparsity constraint on system matrices. In addition, many algorithms neglect the inherent symmetry of noise covariance matrices during the optimization process.

To learn the LDSs with sparse system matrices, we impose the sparsity-promoting prior on them to balance model complexity and modeling error in this paper (Wang et al., 2024). Subsequently, we can combine the likelihood and prior functions to derive the posterior distribution of system matrices following the Bayes' rule. However, directly maximizing such a posterior distribution to estimate system matrices is intractable because the states of LDSs are unknown. To address this issue, we explore the expectation–maximization (EM) algorithm to give an alternate maximum a posteriori (MAP) estimate of hidden states and system matrices. In the expectation step, we use the Rauch–Tung–Striebel (RTS) smoother, also known as the Kalman smoother, to give a closed-

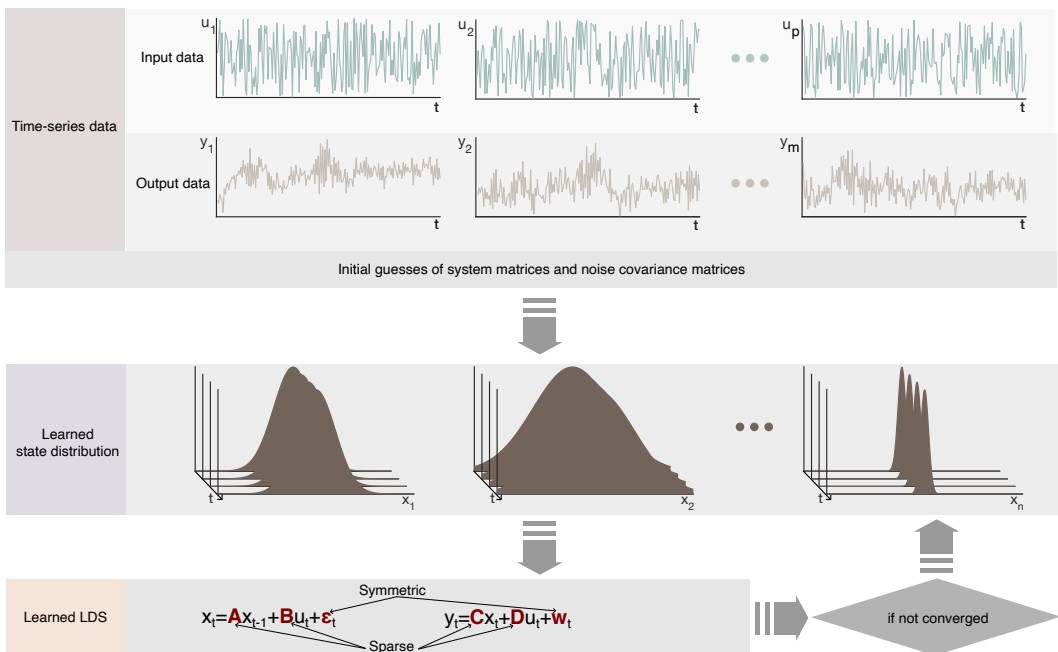

Figure 1: The pipeline of the proposed algorithm. Given time-series data and initial guesses of system matrices and noise covariance matrices, the proposed algorithm alternately learns the hidden state distribution and unknown LDS until it converges.

form update rule for the hidden states. In the maximization step, we leverage the block gradient descent method to optimize the system matrices in turn because they are highly coupled in the objective function. Given the inherent symmetry of noise variance, we consider the derivative rule of structured matrices during the optimization process. By alternately performing the expectation and maximization steps until convergence, the proposed algorithm can determine the sparse system matrices of LDSs from noisy observations.

**Contributions.** The contributions of this paper are summarized as follows:

- Leveraging sparsity-promoting techniques, we propose an algorithm to learn LDSs with sparse system matrices from noisy observations. Particularly, the proposed algorithm gives the MAP estimate of both hidden states and system matrices by exploring the EM algorithm.

- The proposed algorithm utilizes the derivative rule of structured matrices to ensure the symmetry of noise covariance matrices during the optimization process. While many EM-based learning algorithms provide the same update rule for noise covariance matrices, we argue that they use an inappropriate derivative rule.

- Experimental results on simulation and real-world datasets demonstrate that the proposed algorithm outperforms the classical ones on learning LDSs with sparse system matrices.

## 2 RELATED WORK

**Prediction error minimization.** Prediction error minimization (PEM) learns LDSs by minimizing one-step prediction error objective via gradient-based optimization methods (Ljung, 2002; Katayama et al., 2005). Expanding LDSs into linear AutoRegressive Moving Average with eXogenous excitation (ARMAX) or AutoRegressive Moving Average (ARMA) models, Li & Zhang (2006) leverage the PEM-based method to estimate the second-order structural parameters of linear structural systems. Given a symmetric transition matrix, Hazan et al. (2017) present an efficient method for the online prediction of discrete-time LDSs by formulating system identification as an online PEM problem. In particular, Abdalmoaty & Hjalmarsson (2019) extends PEM for learning

stochastic nonlinear models recently. However, PEM is found to be sensitive to initial values and cannot characterize the sparsity of system matrices (Martens, 2010).

**Subspace state-space system identification.** Basically, subspace state-space system identification (4SID) algorithms project data Hankel matrices onto certain subspaces to estimate the extended observability matrix and hidden states using linear algebra tools (Larimore, 1990; Verahegen & Dewilde, 1992; Van Overschee & De Moor, 1994). Leveraging the least squares method, system matrices can thus be recovered from either the extended observability matrix or hidden states (Favoreel et al., 2000). Based on principal component analysis, Wang & Qin (2002) present a new 4SID algorithm to learn LDSs under the errors-in-variables situation. By choosing different weighting matrices to perform the singular value decomposition, Van Overschee & De Moor (2012) provide a geometric framework to unify almost all classical 4SID methods. Further, Huang et al. (2016) presents the Weight-Least-Square method to learn stable LDSs by multiplying the unstable component with a weight matrix. However, 4SID algorithms learn system matrices via the least squares method and thus cannot produce sparse system matrices (Tibshirani, 1996). In particular, it is widely recognized that such algorithms generally cannot obtain accurate system matrices as required (Qin, 2006; Martens, 2010).

**Maximum likelihood estimation.** Because the joint likelihood function of LDSs involves hidden states, the EM algorithm is employed to give the maximum likelihood estimation (MLE) of system matrices (Shumway & Stoffer, 1982; Ghahramani & Hinton, 1996). Leveraging the EM algorithm, the distribution of hidden states can be explicitly derived using the Kalman smoother based on the current estimate of system matrices. Subsequently, it updates system matrices by maximizing the expected log-likelihood with respect to the hidden states. Gibson & Ninness (2005) present a robust MLE of LDSs by implementing the expectation and maximization steps via the LR and Cholesky factorisation respectively. To increase the efficiency of EM for learning LDSs, Martens (2010) proposes an approximate second-order statistics (ASOS) scheme to approximate the expectation step. Combining EM and Lagrangian relaxation, Umenberger et al. (2018) use semidefinite programming to optimize the tight bounds on the likelihood to learn LDSs with model stability constraints. However, such learning algorithms lack the ability to deal with sparse system matrices. Particularly, an inappropriate derivative rule is used to take the derivatives of the likelihood function with respect to noise covariance matrices due to the neglect of the inherent symmetry of them.

## 3 PROBLEM FORMULATION

Generally, LDSs describe time-series data $\{(\boldsymbol{u}_t, \boldsymbol{y}_t)\}_{t=1}^{T}$ through the following stochastic difference equation (Shumway et al., 2000):

$$\boldsymbol{x}_t = \boldsymbol{A}\boldsymbol{x}_{t-1} + \boldsymbol{B}\boldsymbol{u}_t + \boldsymbol{\varepsilon}_t, \tag{1}$$

$$\boldsymbol{y}_t = \boldsymbol{C}\boldsymbol{x}_t + \boldsymbol{D}\boldsymbol{u}_t + \boldsymbol{\omega}_t, \tag{2}$$

where $\boldsymbol{u}_t \in \mathbb{R}^p$ is the input signal, $\boldsymbol{y}_t \in \mathbb{R}^m$ is the noisy observation, $\boldsymbol{x}_t \in \mathbb{R}^n$ is the hidden state, $\boldsymbol{A} \in \mathbb{R}^{n \times n}$, $\boldsymbol{B} \in \mathbb{R}^{n \times p}$, $\boldsymbol{C} \in \mathbb{R}^{m \times n}$, and $\boldsymbol{D} \in \mathbb{R}^{m \times p}$ are the system matrices, and $\boldsymbol{\varepsilon}_t \sim \mathcal{N}(0, \boldsymbol{R})$ and $\boldsymbol{w}_t \sim \mathcal{N}(0, \boldsymbol{Q})$ are the process and measurement noise, respectively. In fact, LDSs are widely used to model complex systems and have been received successful applications in industrial processes (Favoreel et al., 2000). To make predictions about future outputs and unknown states and realize the control of systems, it is necessary to propose an algorithm to learn the model parameters $\boldsymbol{A}$, $\boldsymbol{B}$, $\boldsymbol{C}$, $\boldsymbol{D}$, $\boldsymbol{R}$, and $\boldsymbol{Q}$ from time-series data $\{(\boldsymbol{u}_t, \boldsymbol{y}_t)\}_{t=1}^{T}$. In particular, many systems have sparse topology to enable efficient working mechanisms (Jin et al., 2020a). As a result, the proposed algorithm needs to integrate such priori information into the learning process.

**Sparsity-promoting prior.** Sparsity-promoting priors can enforce the sparsity of model parameters by balancing model complexity and modeling error (Wang et al., 2023; Tripura & Chakraborty, 2023). Because the likelihood function of LDSs is Gaussian distributed, the Student's $t$-distribution prior severing as its conjugate prior can be imposed on each component of the unknown system matrices to promote their sparsity. Generally, the Student's $t$-distribution prior is implemented in a hierarchical way (Tipping, 2001). It imposes a Gaussian prior on the system matrices and then adopt an Inverse-Gamma hyperprior on the unknown variance of the Gaussian distribution. For example, we can impose the Student's $t$-distribution prior on the system matrix $\boldsymbol{A}$ to promote its sparsity as

follows:

$$p(\boldsymbol{A} \mid \boldsymbol{\Gamma}_a) = \prod_{i=1}^{n} \prod_{j=1}^{n} p(\boldsymbol{A}_{ij} \mid \boldsymbol{\Gamma}_{a,ij}) = \prod_{i=1}^{n} \prod_{j=1}^{n} \frac{1}{\sqrt{2\pi \boldsymbol{\Gamma}_{a,ij}}} \exp\left(-\frac{\boldsymbol{A}_{ij}^2}{2\boldsymbol{\Gamma}_{a,ij}}\right), \tag{3}$$

$$p(\boldsymbol{\Gamma}_a) = \prod_{i=1}^{n} \prod_{j=1}^{n} \frac{a_0^{b_0}}{\Gamma(a_0)} \boldsymbol{\Gamma}_{a,ij}^{-a_0-1} \exp\left(-\frac{b_0}{\boldsymbol{\Gamma}_{a,ij}}\right), \tag{4}$$

where $\boldsymbol{A}_{ij}$ and $\boldsymbol{\Gamma}_{a,ij}$ are the $ij$-th components of $\boldsymbol{A}$ and $\boldsymbol{\Gamma}_a$, respectively. To generate non-informative hyperprior on $\boldsymbol{\Gamma}_{a,ij}$, $a_0$ and $b_0$ are typically set to very small values (e.g., $10^{-6}$). In addition, $\boldsymbol{\Gamma}_b$, $\boldsymbol{\Gamma}_c$, $\boldsymbol{\Gamma}_d$, $\boldsymbol{\Gamma}_{b,ij}$, $\boldsymbol{\Gamma}_{c,ij}$, and $\boldsymbol{\Gamma}_{d,ij}$ are defined in a similar manner (see Appendix A).

**Loss function.** Following the Bayes' rule, we can combine the marginal likelihood function and sparsity-promoting prior to estimate the model parameters:

$$p(\boldsymbol{\Theta} \mid \boldsymbol{Y}) \propto \underbrace{p(\boldsymbol{Y} \mid \boldsymbol{\Theta})}_{\text{Marginal likelihood}} \times \underbrace{p(\boldsymbol{\Theta})}_{\text{Prior}}, \tag{5}$$

where $\boldsymbol{Y} = [\boldsymbol{y}_1, \boldsymbol{y}_2, ..., \boldsymbol{y}_T]$ and $\boldsymbol{\Theta} = \{\boldsymbol{A}, \boldsymbol{B}, \boldsymbol{C}, \boldsymbol{D}, \boldsymbol{R}, \boldsymbol{Q}, \boldsymbol{\Gamma}_a, \boldsymbol{\Gamma}_b, \boldsymbol{\Gamma}_c, \boldsymbol{\Gamma}_d\}$. Note that directly maximizing equation 5 is generally intractable because $p(\boldsymbol{Y} \mid \boldsymbol{\Theta})$ is hard to be explicitly computed. However, the EM algorithm provides an iterative optimization framework to address such a problem. Instead of maximizing equation 5, the EM algorithm focuses on iteratively improving the expected value of the log posterior function of $\boldsymbol{\Theta}$ with respect to the hidden state vector $\boldsymbol{X} = [\boldsymbol{x}_1, \boldsymbol{x}_2, ..., \boldsymbol{x}_T]$ as follows:

$$H(\boldsymbol{\Theta} \mid \boldsymbol{\Theta}^k) = \mathbb{E}_{\boldsymbol{X} \sim p(\boldsymbol{X}|\boldsymbol{Y},\boldsymbol{\Theta}^k)}[\log p(\boldsymbol{Y}, \boldsymbol{X} \mid \boldsymbol{\Theta})p(\boldsymbol{\Theta})]. \tag{6}$$

Notably, improving equation 6 is equivalent to improving equation 5 at each iteration (Little & Rubin, 2019).

### 3.1 RAUCH–TUNG–STRIEBEL SMOOTHER

To explicitly compute equation 6, we first need to derive the conditional distribution of $\boldsymbol{x}_t$ given the noisy observation $\boldsymbol{Y}$ and current $\boldsymbol{\Theta}^k = \{\boldsymbol{A}^k, \boldsymbol{B}^k, \boldsymbol{C}^k, \boldsymbol{D}^k, \boldsymbol{R}^k, \boldsymbol{Q}^k, \boldsymbol{\Gamma}_a^k, \boldsymbol{\Gamma}_b^k, \boldsymbol{\Gamma}_c^k, \boldsymbol{\Gamma}_d^k\}$, which can be formulated as a classical smoothing problem. For LDSs, the RTS smoother provides a closed-form smoothing solution for $p(\boldsymbol{x}_t \mid \boldsymbol{Y}, \boldsymbol{\Theta}^k)$.

**Lemma 1.** *(RTS smoother (Särkkä & Svensson, 2023)) For LDSs, the RTS smoother states that*

$$p(\boldsymbol{x}_t \mid \boldsymbol{Y}, \boldsymbol{\Theta}^k) = \mathcal{N}(\boldsymbol{x}_t \mid \boldsymbol{m}_t^k, \boldsymbol{P}_t^k), \tag{7}$$

*where $t = 0, ..., T$. Here, $\boldsymbol{m}_t^k$ and $\boldsymbol{P}_t^k$ are derived via the reverse-time recursions as follows:*

$$\boldsymbol{m}_t^k = \boldsymbol{\mu}_t^k + \boldsymbol{G}_t^k \left(\boldsymbol{m}_{t+1}^k - \overline{\boldsymbol{\mu}}_{t+1}^k\right), \tag{8}$$

$$\boldsymbol{P}_t^k = \boldsymbol{\Sigma}_t^k + \boldsymbol{G}_t^k \left(\boldsymbol{P}_{t+1}^k - \overline{\boldsymbol{\Sigma}}_{t+1}^k\right) (\boldsymbol{G}_t^k)', \tag{9}$$

*with $\boldsymbol{G}_t^k = \boldsymbol{\Sigma}_t^k \left(\boldsymbol{A}^k\right)' \left(\overline{\boldsymbol{\Sigma}}_{t+1}^k\right)^{-1}$. The quantities $\boldsymbol{\mu}_t^k$, $\overline{\boldsymbol{\mu}}_t^k$, $\boldsymbol{\Sigma}_t^k$, and $\overline{\boldsymbol{\Sigma}}_t^k$ coupled in equation 8 and equation 9 are pre-computed using the Kalman filter as follows:*

$$\overline{\boldsymbol{\mu}}_t^k = \boldsymbol{A}^k \boldsymbol{\mu}_{t-1}^k + \boldsymbol{B}^k \boldsymbol{u}_t, \tag{10}$$

$$\overline{\boldsymbol{\Sigma}}_t^k = \boldsymbol{A}^k \boldsymbol{\Sigma}_{t-1}^k \left(\boldsymbol{A}^k\right)' + \boldsymbol{R}^k, \tag{11}$$

$$\boldsymbol{K}_t^k = \overline{\boldsymbol{\Sigma}}_t^k (\boldsymbol{C}^k)' \left(\boldsymbol{C}^k \overline{\boldsymbol{\Sigma}}_t^k (\boldsymbol{C}^k)' + \boldsymbol{Q}^k\right)^{-1}, \tag{12}$$

$$\boldsymbol{\mu}_t^k = \overline{\boldsymbol{\mu}}_t^k + \boldsymbol{K}_t^k \left(\boldsymbol{Y}_t - \boldsymbol{C}^k \overline{\boldsymbol{\mu}}_t^k - \boldsymbol{D}^k \boldsymbol{u}_t\right), \tag{13}$$

$$\boldsymbol{\Sigma}_t^k = \left(\boldsymbol{I}_n - \boldsymbol{K}_t^k \boldsymbol{C}^k\right) \overline{\boldsymbol{\Sigma}}_t^k, \tag{14}$$

*where $\boldsymbol{I}_n$ is an identity matrix of dimension $n$. Note that the reverse-time recursions of equation 8 and equation 9 start from the initial conditions $\boldsymbol{m}_T^k = \boldsymbol{\mu}_T^k$ and $\boldsymbol{P}_T^k = \boldsymbol{\Sigma}_T^k$, and the recursions of equation 10–equation 14 start from the mean $\boldsymbol{\mu}_0^k$ and covariance $\boldsymbol{\Sigma}_0^k$ of the initial state $\boldsymbol{x}_0$.*

Besides $p(\boldsymbol{x}_t \mid \boldsymbol{Y}, \boldsymbol{\Theta}^k)$, we also need to derive the covariance matrix between the adjacent states $\boldsymbol{x}_t$ and $\boldsymbol{x}_{t-1}$ given $\boldsymbol{Y}$ and $\boldsymbol{\Theta}^k$ to compute equation 6. To address this issue, the following lemma gives necessary recursions.

**Lemma 2.** *(The lag-one covariance smoother (Särkkä & Svensson, 2023)) For the LDSs, the covariance matrix $\boldsymbol{P}_{t,t-1}^k$ between the adjacent states $\boldsymbol{x}_t$ and $\boldsymbol{x}_{t-1}$ given $\boldsymbol{Y}$ and $\boldsymbol{\Theta}^k$ can be recursively derived as follows:*

$$\boldsymbol{P}_{t,t-1}^k = \boldsymbol{\Sigma}_t^k (\boldsymbol{G}_{t-1}^k)' + \boldsymbol{G}_t^k \left( \boldsymbol{P}_{t+1,t}^k - \boldsymbol{A}^k \boldsymbol{\Sigma}_t^k \right) (\boldsymbol{G}_{t-1}^k)' \tag{15}$$

*with the initial condition $\boldsymbol{P}_{T,T-1}^k = \left( \boldsymbol{I}_n - \boldsymbol{K}_T^k \boldsymbol{C}^k \right) \boldsymbol{A}^k \boldsymbol{\Sigma}_{T-1}^k$.*

Based on Lemmas 1 and 2, we are able to calculate the loss function in equation 6 as follows:

$$H(\boldsymbol{\Theta} \mid \boldsymbol{\Theta}^k) = H_1(\boldsymbol{A}, \boldsymbol{B}, \boldsymbol{R}) + H_2(\boldsymbol{C}, \boldsymbol{D}, \boldsymbol{Q}) + H_3(\boldsymbol{A}, \boldsymbol{B}, \boldsymbol{C}, \boldsymbol{D}, \boldsymbol{\Gamma}_a, \boldsymbol{\Gamma}_b, \boldsymbol{\Gamma}_c, \boldsymbol{\Gamma}_d), \tag{16}$$

where

$$H_1(\boldsymbol{A}, \boldsymbol{B}, \boldsymbol{R}) = \mathbb{E}_{\boldsymbol{X} \sim p(\boldsymbol{X} \mid \boldsymbol{Y}, \boldsymbol{\Theta}^k)}[\log p(\boldsymbol{X} \mid \boldsymbol{A}, \boldsymbol{B}, \boldsymbol{R})], \tag{17}$$

$$H_2(\boldsymbol{C}, \boldsymbol{D}, \boldsymbol{Q}) = \mathbb{E}_{\boldsymbol{X} \sim p(\boldsymbol{X} \mid \boldsymbol{Y}, \boldsymbol{\Theta}^k)}[\log p(\boldsymbol{Y} \mid \boldsymbol{X}, \boldsymbol{C}, \boldsymbol{D}, \boldsymbol{Q})], \tag{18}$$

$$H_3(\boldsymbol{A}, \boldsymbol{B}, \boldsymbol{C}, \boldsymbol{D}, \boldsymbol{\Gamma}_a, \boldsymbol{\Gamma}_b, \boldsymbol{\Gamma}_c, \boldsymbol{\Gamma}_d) = \mathbb{E}_{\boldsymbol{X} \sim p(\boldsymbol{X} \mid \boldsymbol{Y}, \boldsymbol{\Theta}^k)}[\log p(\boldsymbol{A}, \boldsymbol{\Gamma}_a)p(\boldsymbol{B}, \boldsymbol{\Gamma}_b)p(\boldsymbol{C}, \boldsymbol{\Gamma}_c)p(\boldsymbol{D}, \boldsymbol{\Gamma}_d)]. \tag{19}$$

Due to the limited space, the detailed derivation of equation 16 and explicit mathematical expressions of $H_1(\boldsymbol{A}, \boldsymbol{B}, \boldsymbol{R})$, $H_2(\boldsymbol{C}, \boldsymbol{D}, \boldsymbol{Q})$, and $H_3(\boldsymbol{A}, \boldsymbol{B}, \boldsymbol{C}, \boldsymbol{D}, \boldsymbol{\Gamma}_a, \boldsymbol{\Gamma}_b, \boldsymbol{\Gamma}_c, \boldsymbol{\Gamma}_d)$ are given in Appendix B.1.

## 3.2 PARAMETER AND HYPERPARAMETER LEARNING

As $H(\boldsymbol{\Theta} \mid \boldsymbol{\Theta}^k)$ is a non-convex function and unknown parameters are highly coupled, it is difficult to obtain an efficient algorithm with theoretical guarantees for solving such a problem. A heuristic method is to leverage the block gradient descent method to iteratively optimize the model parameters.

**Update procedures of $\boldsymbol{A}$, $\boldsymbol{B}$, $\boldsymbol{C}$, and $\boldsymbol{D}$.** For MLE, leveraging the EM algorithm can give a closed-form solution to update $\boldsymbol{A}$, $\boldsymbol{B}$, $\boldsymbol{C}$, and $\boldsymbol{D}$ (Ghahramani & Hinton, 1996; Gibson & Ninness, 2005). However, it is intractable to obtain a similar update procedure in this case due to the introduction of the sparsity-promoting prior. For example, we can calculate the derivative of $H(\boldsymbol{\Theta} \mid \boldsymbol{\Theta}^k)$ with respect to $\boldsymbol{A}$ at the $k$th iteration as follows:

$$\frac{\partial H_1(\boldsymbol{A}, \boldsymbol{B}^k, \boldsymbol{R}^k)}{\partial \boldsymbol{A}} + \frac{\partial H_3(\boldsymbol{A}, \boldsymbol{B}^k, \boldsymbol{C}^k, \boldsymbol{D}^k, \boldsymbol{\Gamma}_a^k, \boldsymbol{\Gamma}_b^k, \boldsymbol{\Gamma}_c^k, \boldsymbol{\Gamma}_d^k)}{\partial \boldsymbol{A}}$$

$$= \sum_{t=1}^T \left( \boldsymbol{R}^k \right)^{-1} \left( \boldsymbol{P}_{t,t-1}^k + \left( \boldsymbol{m}_t^k - \boldsymbol{A} \boldsymbol{m}_{t-1}^k - \boldsymbol{B}^k \boldsymbol{u}_t \right) (\boldsymbol{m}_{t-1}^k)' - \boldsymbol{A} \boldsymbol{P}_{t-1}^k \right) - \boldsymbol{A} \odot \overline{\boldsymbol{\Gamma}}_a^k, \tag{20}$$

where the $ij$th component of $\overline{\boldsymbol{\Gamma}}_a^k$ is $1/\boldsymbol{\Gamma}_{a,ij}^k$ and $\odot$ is the Hadamard product. Obviously, setting equation 20 to zero and solving for $\boldsymbol{A}$ cannot give a closed-form solution. To address this issue, we approximate $\boldsymbol{R}^k$ using the diagonal matrix formed by its diagonal components to facilitate the optimization process. As such, we can calculate the derivative of $H(\boldsymbol{\Theta} \mid \boldsymbol{\Theta}^k)$ with respect to the $r$th row of $\boldsymbol{A}$, denoted as $\boldsymbol{A}_r$, at the $k$th iteration as follows:

$$\frac{\partial H_1(\boldsymbol{A}, \boldsymbol{B}^k, \boldsymbol{R}^k)}{\partial \boldsymbol{A}_r} + \frac{\partial H_3(\boldsymbol{A}, \boldsymbol{B}^k, \boldsymbol{C}^k, \boldsymbol{D}^k, \boldsymbol{\Gamma}_a^k, \boldsymbol{\Gamma}_b^k, \boldsymbol{\Gamma}_c^k, \boldsymbol{\Gamma}_d^k)}{\partial \boldsymbol{A}_r}$$

$$= \sum_{t=1}^T \left( \boldsymbol{R}_{rr}^k \right)^{-1} \left( \boldsymbol{P}_{t,t-1,r}^k + (\boldsymbol{m}_{t,r}^k - \boldsymbol{A}_r \boldsymbol{m}_{t-1}^k - \boldsymbol{B}^k \boldsymbol{u}_t) \left( \boldsymbol{m}_{t-1}^k \right)' - \boldsymbol{A}_r \boldsymbol{P}_{t-1}^k \right) - \boldsymbol{A}_r \overline{\boldsymbol{\Gamma}}_{a,r}^{kd}, \tag{21}$$

where $\boldsymbol{R}_{rr}^k$ is the $rr$th component of $\boldsymbol{R}^k$, $\boldsymbol{P}_{t,t-1,r}^k$, $\boldsymbol{m}_{t,r}^k$, and $\boldsymbol{B}_r^k$ are the $r$th rows of $\boldsymbol{P}_{t,t-1}^k$, $\boldsymbol{m}_t^k$, and $\boldsymbol{B}^k$, respectively. In particular, $\overline{\boldsymbol{\Gamma}}_{a,r}^{kd} = \text{diag}[\overline{\boldsymbol{\Gamma}}_{a,r}^k]$ with $\overline{\boldsymbol{\Gamma}}_{a,r}^k$ being the $r$th row of $\overline{\boldsymbol{\Gamma}}_a^k$. Set-

ting equation 21 to zero leads to

$$A_r^{k+1} = \left( \sum_{t=1}^T \left( (m_{t,r}^k - B_r^k u_t)(m_{t-1}^k)' + P_{t,t-1,r}^k \right) \right)$$

$$\times \left( \sum_{t=1}^T \left( P_{t-1}^k + m_{t-1}^k \left( m_{t-1}^k \right)' \right) + R_{rr}^k \overline{\Gamma}_{a,r}^{kd} \right)^{-1}. \tag{22}$$

The detailed derivation of equation 22 can be found in Appendix B.2. Similarly, we can update the $r$th row of $B$, $C$, and $D$ as follows:

$$B_r^{k+1} = \left( \sum_{t=1}^T \left( m_{t,r}^k - A_r^{k+1} m_{t-1}^k \right) u_t' \right) \left( \sum_{t=1}^T u_t u_t' + R_{rr}^k \overline{\Gamma}_{b,r}^{kd} \right)^{-1}, \tag{23}$$

$$C_r^{k+1} = \left( \sum_{t=1}^T \left( y_{t,r} - D_r^k u_t \right) (m_t^k)' \right) \left( \sum_{t=1}^T \left( P_t^k + m_t^k (m_t^k)' \right) + Q_{rr}^k \overline{\Gamma}_{c,r}^{kd} \right)^{-1}, \tag{24}$$

$$D_r^{k+1} = \left( \sum_{t=1}^T \left( y_{t,r} - C_r^{k+1} m_t^k \right) u_t' \right) \left( \sum_{t=1}^T u_t u_t' + Q_{rr}^k \overline{\Gamma}_{d,r}^{kd} \right)^{-1}, \tag{25}$$

where $y_{t,r}$ is the $r$th component of $y_t$, $Q_{rr}^k$ is the $rr$th component of $Q^k$, and $\overline{\Gamma}_{b,r}^{kd}$, $\overline{\Gamma}_{c,r}^{kd}$, and $\overline{\Gamma}_{d,r}^{kd}$ are defined as that of $\overline{\Gamma}_{a,r}^{kd}$.

**Update procedures of $R$ and $Q$.** Because many learning algorithms do not consider the inherent symmetry of noise covariance matrices, we argue that they use an inappropriate derivative rule to calculate the derivatives of the loss function with respect to $R$ and $Q$ (Gibson & Ninness, 2005; Umenberger et al., 2018). Based on the derivative rule of structured matrices (Petersen et al., 2008), we can calculate the derivative of $H(\Theta \mid \Theta^k)$ with respect to $R$ at the $k$th iteration as follows:

$$\frac{\partial H_1(A^{k+1}, B^{k+1}, R)}{\partial R} = \frac{2L(R) - L(R) \odot I_n}{2}, \tag{26}$$

where

$$L(R) = \sum_{t=1}^T R^{-1} \left( m_t^k - A^{k+1} m_{t-1}^k - B^{k+1} u_t \right) \left( m_t^k - A^{k+1} m_{t-1}^k - B^{k+1} u_t \right)' R^{-1}$$

$$+ \sum_{t=1}^T R^{-1} \left( P_t^k - A^{k+1} P_{t,t-1}^k - P_{t,t-1}^k \left( A^{k+1} \right)' + A^{k+1} P_{t-1}^k \left( A^{k+1} \right)' \right) R^{-1} - T R^{-1}. \tag{27}$$

To update $R$, we first introduce the following lemma to simplify the derivation.

**Lemma 3.** *For a square matrix $H \in \mathbb{R}^{n \times n}$, if $2H - H \odot I_n = 0$, we have $H = 0$.*

The proof of Lemma 3 is straightforward and thus is omitted here. Based on Lemma 3, setting $L(R)$ to zero yields

$$R^{k+1} = \frac{\sum_{t=1}^T \left( m_t^k - A^{k+1} m_{t-1}^k - B^{k+1} u_t \right) \left( m_t^k - A^{k+1} m_{t-1}^k - B^{k+1} u_t \right)'}{T}$$

$$+ \frac{\sum_{t=1}^T \left( P_t^k - A^{k+1} P_{t,t-1}^k - P_{t,t-1}^k (A^{k+1})' + A^{k+1} P_{t-1}^k (A^{k+1})' \right)}{T}. \tag{28}$$

**Remark 1.** *Without considering the symmetry of $R$, the derivative of $H(\Theta \mid \Theta^k)$ with respect to $R$ is equal to $L(R)$. Hence, many learning algorithms give the same update procedure of $R$ as ours. However, we argue that they use an inappropriate derivative rule during the optimization process.*

Similarly, we can update $Q$ as follows:

$$Q^{k+1} = \frac{\sum_{t=1}^T \left( (y_t - C^{k+1} m_t^k - D^{k+1} u_t)(y_t - C^{k+1} m_t^k - D^{k+1} u_t)' + C^{k+1} P_t^k (C^{k+1})' \right)}{T}. \tag{29}$$

**Update procedures of $\boldsymbol{\Gamma}_a$, $\boldsymbol{\Gamma}_b$, $\boldsymbol{\Gamma}_c$, and $\boldsymbol{\Gamma}_d$.** Because each component of $\boldsymbol{\Gamma}_a$, $\boldsymbol{\Gamma}_b$, $\boldsymbol{\Gamma}_c$, and $\boldsymbol{\Gamma}_d$ is independent, we can update them individually. For example, we can calculate the derivative of $H(\boldsymbol{\Theta} \mid \boldsymbol{\Theta}^k)$ with respect to $\boldsymbol{\Gamma}_{a,ij}$ at the $k$th iteration as follows:

$$\frac{H_3(\boldsymbol{A}^{k+1}, \boldsymbol{B}^{k+1}, \boldsymbol{C}^{k+1}, \boldsymbol{D}^{k+1}, \boldsymbol{\Gamma}_a, \boldsymbol{\Gamma}_b^k, \boldsymbol{\Gamma}_c^k, \boldsymbol{\Gamma}_d^k)}{\partial \boldsymbol{\Gamma}_{a,ij}} = -\frac{2a_0 + 3}{2\boldsymbol{\Gamma}_{a,ij}} + \frac{(\boldsymbol{A}_{ij}^{k+1})^2 + 2b_0}{2\boldsymbol{\Gamma}_{a,ij}^2}. \tag{30}$$

Setting equation 30 to zero and solving for $\boldsymbol{\Gamma}_{a,ij}$ leads to:

$$\boldsymbol{\Gamma}_{a,ij}^{k+1} = \frac{(\boldsymbol{A}_{ij}^{k+1})^2 + 2b_0}{2a_0 + 3}. \tag{31}$$

Similarly, we can update each component of $\boldsymbol{\Gamma}_b$, $\boldsymbol{\Gamma}_c$, and $\boldsymbol{\Gamma}_d$ as follows:

$$\boldsymbol{\Gamma}_{b,ij}^{k+1} = \frac{(\boldsymbol{B}_{ij}^{k+1})^2 + 2b_0}{2a_0 + 3}, \tag{32}$$

$$\boldsymbol{\Gamma}_{c,ij}^{k+1} = \frac{(\boldsymbol{C}_{ij}^{k+1})^2 + 2b_0}{2a_0 + 3}, \tag{33}$$

$$\boldsymbol{\Gamma}_{d,ij}^{k+1} = \frac{(\boldsymbol{D}_{ij}^{k+1})^2 + 2b_0}{2a_0 + 3}. \tag{34}$$

Based on the block gradient descent method, we derive an analytical update procedure for learning $\boldsymbol{\Theta}$. During the optimization process of the system matrices, we use diagonal matrices to approximate $\boldsymbol{R}$ and $\boldsymbol{Q}$ to give a closed-form update rule for them. During the optimization process of $\boldsymbol{R}$ and $\boldsymbol{Q}$, we employ the derivative rule of structured matrices to ensure their symmetry. Experimental results demonstrate that such a learning algorithm can learn LDSs with sparse system matrices accurately. Finally, Algorithm 1 summarizes the procedure for learning LDSs with sparse system matrices.

**Remark 2.** *For LDSs without input signals, the proposed method can also learn LDSs with sparse system matrices from $\{\boldsymbol{y}_t\}_{t=1}^T$ in an unsupervised manner by simply removing $\boldsymbol{B}$, $\boldsymbol{D}$, and $\boldsymbol{u}_t$ from the related update procedures or directly setting them to zero in the optimization process.*

---

**Algorithm 1:** The proposed learning algorithm for LDSs

---

**Input:** Time-series data $\{(\boldsymbol{u}_t, \boldsymbol{y}_t)\}_{t=1}^T$, initial guess of $\boldsymbol{\Theta}$ and maximum number of iterations $k_{max}$
**Output:** MAP estimate of $\boldsymbol{\Theta}$ and $\{\boldsymbol{x}_t\}_{t=1}^T$

1 **for** $k = 1, ..., k_{max}$ **do**
2     // MAP estimate of $\{\boldsymbol{x}_t\}_{t=1}^T$
3     **for** $t = 1, ..., T$ **do**
4         Update the mean $\boldsymbol{\mu}_t^k$ of $\boldsymbol{x}_t$ via equation 13 ;
5         Update the variance $\boldsymbol{\Sigma}_t^k$ of $\boldsymbol{x}_t$ via equation 14 ;
6         Update the covariance $\boldsymbol{P}_{t,t-1}^k$ between $\boldsymbol{x}_t$ and $\boldsymbol{x}_{t-1}$ via equation 15 ;
7     **end**
8     // MAP estimate of $\boldsymbol{\Theta}$
9     Update system matrices $\boldsymbol{A}$, $\boldsymbol{B}$, $\boldsymbol{C}$, and $\boldsymbol{D}$ via equation 22– equation 25, respectively;
10     Update noise covariance matrices $\boldsymbol{R}$ and $\boldsymbol{Q}$ via equation 28 and equation 29, respectively;
11     Update hyperparameter matrices $\boldsymbol{\Gamma}_a$, $\boldsymbol{\Gamma}_b$, $\boldsymbol{\Gamma}_c$, and $\boldsymbol{\Gamma}_d$ via equation 31– equation 34, respectively;
12     **if** *a stopping criterion is satisfied* **then**
13         Break;
14     **end**
15 **end**

---

## 4 SIMILARITY TRANSFORMATION OF LDSS

For LDSs, the similarity transformation is an important mathematical operation to transform them into different coordinate systems, making it easier to analyze system properties like controllability,

observability, and stability. Specifically, we can transform the state vector $\boldsymbol{x}_t$ into a new state vector $\overline{\boldsymbol{x}}_t$ through the relation:

$$\overline{\boldsymbol{x}}_t = \boldsymbol{P}\boldsymbol{x}_t, \tag{35}$$

where $\boldsymbol{P} \in \mathbb{R}^{n \times n}$ is a nonsingular matrix. As such, we can derive an equivalent realization of the original LDSs as follows (see Appendix C):

$$\overline{\boldsymbol{x}}_t = \overline{\boldsymbol{A}}\overline{\boldsymbol{x}}_{t-1} + \overline{\boldsymbol{B}}\boldsymbol{u}_t + \overline{\boldsymbol{\varepsilon}}_t, \tag{36}$$

$$\boldsymbol{y}_t = \overline{\boldsymbol{C}}\overline{\boldsymbol{x}}_t + \boldsymbol{D}\boldsymbol{u}_t + \boldsymbol{\omega}_t, \tag{37}$$

where $\overline{\boldsymbol{A}} = \boldsymbol{P}\boldsymbol{A}\boldsymbol{P}^{-1}, \overline{\boldsymbol{B}} = \boldsymbol{P}\boldsymbol{B}, \overline{\boldsymbol{C}} = \boldsymbol{C}\boldsymbol{P}^{-1}$, and $\overline{\boldsymbol{\varepsilon}}_t \sim \mathcal{N}(0, \boldsymbol{P}\boldsymbol{R}\boldsymbol{P}')$. However, the similarity transformation makes it particularly difficult to accurately learn system matrices. Given the input signals $\{\boldsymbol{u}_t\}_{t=1}^T$, the transformed LDSs can produce the same output data $\{\boldsymbol{y}_t\}_{t=1}^T$ as that of the original LDSs. Hence, classical learning algorithms for LDSs only learn the system matrices up to a similar transformation (Viberg, 1994). For LDSs with sparse system matrices, such a transformation changes not only the values but, more importantly, the topological structure of the system matrices, resulting in misinterpretation of intrinsic working mechanisms.

## 4.1 BENEFIT OF SPARSE-PROMOTING PRIORS

Unlike classical learning algorithms, the proposed algorithm learns LDSs with sparse system matrices by adopting a sparsity-promoting prior to balance model complexity and modeling error. Given the sparsity constraint of system matrices, the similarity transformation cannot be applied using any arbitrary nonsingular matrix. For the LDSs with sparse system matrices following the *Occam's razor* principle, the nonsingular matrix is typically restricted to be a generalized permutation matrix; otherwise, the transformed LDSs will include redundant parameters to describe the systems. For example, if we consider the LDSs with sparse system matrices as follows:

$$\boldsymbol{A} = \begin{bmatrix} 0 & 0.9 \\ 0.9 & 0 \end{bmatrix}, \boldsymbol{B} = \begin{bmatrix} 2 & 0 \\ 0 & 2 \end{bmatrix}, \boldsymbol{C} = \begin{bmatrix} 2 & 0 \\ 0 & 2 \end{bmatrix}, \boldsymbol{D} = \begin{bmatrix} 1.5 & 0 \\ 0 & 1.5 \end{bmatrix}, \tag{38}$$

it is easy to verify that such a system follows the *Occam's razor* principle because the rank of system matrices is equal to the number of nonzero components. Hence, we can derive the nonsingular matrix $\boldsymbol{P}$ must satisfies

$$\boldsymbol{P} = \begin{bmatrix} a & 0 \\ 0 & b \end{bmatrix} \quad \text{or} \quad \boldsymbol{P} = \begin{bmatrix} 0 & a \\ b & 0 \end{bmatrix}, \tag{39}$$

where $a$ and $b$ are arbitrary constants. As such, the transformed system matrices do not introduce additional parameters to increase model complexity.

Note that applying the similarity transformation with a generalized permutation matrix to the original state variables will scale their magnitudes and reorder them. However, it will scale the nonzero components and permute the rows or columns of system matrices accordingly. Hence, an additional advantage of the sparse-promoting prior is its ability to maximally preserve the inherent topological structure among the variables. While the learned system matrices differ from the true ones in scale, such a difference is only caused by the scaled definition of state variables. Hence, the learned LDS has the same topological structure and dynamic behavior as the real one.

## 5 EXPERIMENT

In this section, we validate the proposed algorithm on simulation and real-world datasets. In addition, we compare the proposed algorithm with classical ones mentioned previously to demonstrate its superior performance, including PEM, 4SID, and MLE. Here, we use the built-in functions **n4sid** and **pem** of Matlab to implement the 4SID and PEM algorithms, respectively. To implement MLE, we remove the sparsity-promoting priors from our derivation and revise the code accordingly. In all experiments, the dataset is split into training and testing sets with a 2:1 ratio, where 66.7% of the data is used for training and 33.3% for testing. Here, we use the mean relative error (MRE) to evaluate the performance of all the algorithms defined as follows:

$$\text{MRE} = \sum_{t=1}^T \frac{\|\boldsymbol{y}_t - \hat{\boldsymbol{y}}_t\|_2^2}{T\|\boldsymbol{y}_t\|_2^2}, \tag{40}$$

Table 1: Learned result of all the algorithms on the simulation system

| Method | Ours | PEM | 4SID | MLE |
|---|---|---|---|---|
| $A$ | $\begin{bmatrix} 0 & 0.900 \\ 0.897 & 0 \end{bmatrix}$ | $\begin{bmatrix} 0 & 0.877 \\ 0.917 & -0.002 \end{bmatrix}$ | $\begin{bmatrix} 0.897 & 0.023 \\ 0.038 & -0.898 \end{bmatrix}$ | $\begin{bmatrix} 0.007 & 0.889 \\ 0.905 & -0.009 \end{bmatrix}$ |
| $B$ | $\begin{bmatrix} 4.004 & 0 \\ 0 & 4.024 \end{bmatrix}$ | $\begin{bmatrix} -14.284 & 16.858 \\ 16.979 & -14.985 \end{bmatrix}$ | $\begin{bmatrix} 0.005 & 0.004 \\ -0.001 & 0 \end{bmatrix}$ | $\begin{bmatrix} 2.887 & 1.235 \\ 1.308 & 2.848 \end{bmatrix}$ |
| $C$ | $\begin{bmatrix} 1.001 & 0 \\ 0 & 1.011 \end{bmatrix}$ | $\begin{bmatrix} 0.880 & 0.745 \\ 0.766 & 0.861 \end{bmatrix}$ | $\begin{bmatrix} 422.168 & -170.034 \\ 415.973 & 179.648 \end{bmatrix}$ | $\begin{bmatrix} 1.777 & -0.778 \\ -0.760 & 1.763 \end{bmatrix}$ |
| $D$ | $\begin{bmatrix} 1.473 & 0 \\ 0 & 1.480 \end{bmatrix}$ | $\begin{bmatrix} 5.521 & 0.099 \\ 0.017 & 5.550 \end{bmatrix}$ | $\begin{bmatrix} 0 & 0 \\ 0 & 0 \end{bmatrix}$ | $\begin{bmatrix} 1.404 & 0.113 \\ -0.092 & 1.468 \end{bmatrix}$ |
| $R$ | $\begin{bmatrix} 1.879 & 0.986 \\ 0.986 & 1.877 \end{bmatrix}$ | $\begin{bmatrix} - & - \\ - & - \end{bmatrix}$ | $\begin{bmatrix} - & - \\ - & - \end{bmatrix}$ | $\begin{bmatrix} 1.543 & 1.402 \\ 1.402 & 1.546 \end{bmatrix}$ |
| $Q$ | $\begin{bmatrix} 0.473 & 0.199 \\ 0.199 & 0.481 \end{bmatrix}$ | $\begin{bmatrix} - & - \\ - & - \end{bmatrix}$ | $\begin{bmatrix} - & - \\ - & - \end{bmatrix}$ | $\begin{bmatrix} 0.455 & 0.191 \\ 0.191 & 0.469 \end{bmatrix}$ |
| MRE | **7.35%** | 16.37% | 17.25% | 7.38% |

where $\{\hat{y}_t\}_{t=1}^T$ is the sequence of data points generated by the learned systems in response to the same input signals. Experimental results illustrate that the proposed algorithm outperforms classical ones on learning LDSs with sparse system matrices.

## 5.1 A SYNTHETIC SYSTEM FOLLOWING THE OCCAM'S RAZOR PRINCIPLE

First, we consider a synthetic system to facilitate the comparison between the proposed algorithm and classical ones as follows:

$$\boldsymbol{x}_t = \begin{bmatrix} 0 & 0.9 \\ 0.9 & 0 \end{bmatrix} \boldsymbol{x}_{t-1} + \begin{bmatrix} 2 & 0 \\ 0 & 2 \end{bmatrix} \boldsymbol{u}_t + \boldsymbol{\varepsilon}_t, \quad \boldsymbol{\varepsilon}_t \sim \left( \begin{bmatrix} 0 \\ 0 \end{bmatrix}, \begin{bmatrix} 0.49 & 0.25 \\ 0.25 & 0.49 \end{bmatrix} \right), \tag{41}$$

$$\boldsymbol{y}_t = \begin{bmatrix} 2 & 0 \\ 0 & 2 \end{bmatrix} \boldsymbol{x}_t + \begin{bmatrix} 1.5 & 0 \\ 0 & 1.5 \end{bmatrix} \boldsymbol{u}_t + \boldsymbol{w}_t, \quad \boldsymbol{w}_t \sim \left( \begin{bmatrix} 0 \\ 0 \end{bmatrix}, \begin{bmatrix} 0.49 & 0.25 \\ 0.25 & 0.49 \end{bmatrix} \right). \tag{42}$$

To generate data points, the initial values of $\boldsymbol{x}_0$ are drawn from the Gaussian distribution with mean $[1, 1]'$ and an identity matrix as the covariance, and the input signal $\boldsymbol{u}_t$ is drawn from the uniform distribution on $[0, 2]$. As for algorithm implementation, we collect 2000 data points and set the initial value of $\boldsymbol{A}, \boldsymbol{B}, \boldsymbol{C}, \boldsymbol{D}, \boldsymbol{R}$, and $\boldsymbol{Q}$ both to be an identity matrix. The learned parameters less than the threshold 0.001 are removed from the result. In Table 1, we give the learned system matrices of all the algorithms and the corresponding MRE.

Because **n4sid** and **pem** consider the innovation representation of the LDSs (Qin, 2006), they focus on learning the innovation covariance matrix instead of $\boldsymbol{R}$ and $\boldsymbol{Q}$, which are thus omitted here. Obviously, the learned system matrices of classical algorithms are completely different with the original ones, making it difficult for us to understand the system. However, sparse-promoting priors will restrict the nonsingular matrix $\boldsymbol{P}$ of the similarity transformation to be a generalized permutation matrix for this system. Comparing the learned $\boldsymbol{B}$ and $\boldsymbol{C}$ of the proposed algorithm with the real ones, we can derive $\boldsymbol{P} \approx 2\boldsymbol{I}_2$, which is indeed consistent with the theoretical analysis. Hence, the learned LDS of the proposed algorithm preserves the topological structure of the system, differing only in the scale of the parameters. Consequently, the learned LDS can enable us to explore the working mechanisms of the system.

## 5.2 INDUSTRIAL PROCESS SYSTEMS

Next, we validate the proposed algorithm on the real-world datasets obtained from the Database for the Identification of Systems, which are standard datasets used for learning LDSs (Zhu et al., 1994; Martens, 2010).

Table 2: Learned result of all the algorithms on the industrial process systems

| Dataset | Industrial evaporator | | | | Glass furnace | | | |
|---|---|---|---|---|---|---|---|---|
| Method | Ours | PEM | 4SID | MLE | Ours | PEM | 4SID | MLE |
| MRE | **13.74%** | 17.90% | 43.77% | 18.00% | **18.74%** | 62.47% | 24.32% | 30.27% |

**Industrial evaporatoration systems.** In industry, multiple-stage evaporators are widely used to reduce the water content of a product such as milk. The dataset is composed of 3-dimensional time-series with a length of 6305. The inputs consist of the feed flow, vapor flow to the first evaporator stage, and cooling water flow to the condenser, while the outputs include the dry matter content, flow rate, and temperature of the product.

**Glass furnaces.** The second dataset comes from the Philips glass furnace, which is used to melt raw materials into glass. The glass furnace has two burners and one ventilator. Hence, the dataset includes two heating inputs and one cooling input with a length of 1247. In addition, we collect three outputs from temperature sensors in a cross section of the furnace.

Table 2 displays the MRE between the predicted outputs of all the learned LDSs and real ones. Due to the lack of the ground truth, the learned system matrices of all the algorithms are not depicted for comparison. Note that the proposed algorithm obtains minimum MRE on both datasets, demonstrating its superiority over classical algorithms.

## 6 DISCUSSION

To learn the LDSs with sparse system matrices, we impose sparsity-promoting priors on system matrices to balance model complexity and modeling error in this paper. Following the MAP principle, we then learn system matrices by exploring the EM algorithm to maximize the loss function composed of the priors and likelihood function. During the optimization process, we use the derivative rule of structured matrices to ensure the symmetry of noise covariance matrices. In addition, we find that the sparsity-promoting prior is capable of retaining the topological structure of the LDSs, as the nonsingular matrix of the similarity transformation is typically limited to be a generalized permutation matrix. Hence, the proposed algorithm is more useful for us to explore the interacting laws of the LDSs compared to the classical ones.

There still remains some potential limitations for the proposed algorithm. First, it cannot determine the order $n$ of the system from data directly. While we can compare the performance of the learned systems across different orders to select the best one, such a method is quite exhaustive. The other limitation is that the similarity transformation may shrink many parameters to very small values, potentially leading to numerical errors. However, we believe that the proposed algorithm sheds a light on the learning of LDSs with sparse system matrices. In our future work, we hope to explore how to exactly learn LDSs with additional constraints.

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

# APPENDIX

## A  SPARSITY-PROMOTING PRIOR

Besides $\boldsymbol{A}$, we also impose the sparsity-promoting priors on $\boldsymbol{B}, \boldsymbol{C}$, and $\boldsymbol{D}$ as follows:

$$p(\boldsymbol{B} \mid \boldsymbol{\Gamma}_b) = \prod_{i=1}^{n} \prod_{j=1}^{p} p(\boldsymbol{B}_{ij} \mid \boldsymbol{\Gamma}_{b,ij}) = \prod_{i=1}^{n} \prod_{j=1}^{p} \frac{1}{\sqrt{2\pi\boldsymbol{\Gamma}_{b,ij}}} \exp\left(-\frac{\boldsymbol{B}_{ij}^2}{2\boldsymbol{\Gamma}_{b,ij}}\right), \tag{43}$$

$$p(\boldsymbol{C} \mid \boldsymbol{\Gamma}_c) = \prod_{i=1}^{m} \prod_{j=1}^{n} p(\boldsymbol{C}_{ij} \mid \boldsymbol{\Gamma}_{c,ij}) = \prod_{i=1}^{m} \prod_{j=1}^{n} \frac{1}{\sqrt{2\pi\boldsymbol{\Gamma}_{c,ij}}} \exp\left(-\frac{\boldsymbol{C}_{ij}^2}{2\boldsymbol{\Gamma}_{c,ij}}\right), \tag{44}$$

$$p(\boldsymbol{D} \mid \boldsymbol{\Gamma}_d) = \prod_{i=1}^{m} \prod_{j=1}^{p} p(\boldsymbol{D}_{ij} \mid \boldsymbol{\Gamma}_{d,ij}) = \prod_{i=1}^{m} \prod_{j=1}^{p} \frac{1}{\sqrt{2\pi\boldsymbol{\Gamma}_{d,ij}}} \exp\left(-\frac{\boldsymbol{D}_{ij}^2}{2\boldsymbol{\Gamma}_{d,ij}}\right), \tag{45}$$

where $\boldsymbol{\Gamma}_{b,ij}, \boldsymbol{\Gamma}_{c,ij}$, and $\boldsymbol{\Gamma}_{d,ij}$ are the $ij$th component of $\boldsymbol{\Gamma}_b, \boldsymbol{\Gamma}_c$, and $\boldsymbol{\Gamma}_d$, respectively. To complete the hierarchy, the Inverse-Gamma distribution prior is imposed on each component of $\boldsymbol{\Gamma}_b, \boldsymbol{\Gamma}_c$, and $\boldsymbol{\Gamma}_d$ as follows:

$$p(\boldsymbol{\Gamma}_b) = \prod_{i=1}^{n} \prod_{j=1}^{p} \frac{a_0^{b_0}}{\Gamma(a_0)} \boldsymbol{\Gamma}_{b,ij}^{-a_0-1} \exp\left(-\frac{b_0}{\boldsymbol{\Gamma}_{b,ij}}\right), \tag{46}$$

$$p(\boldsymbol{\Gamma}_c) = \prod_{i=1}^{m} \prod_{j=1}^{n} \frac{a_0^{b_0}}{\Gamma(a_0)} \boldsymbol{\Gamma}_{c,ij}^{-a_0-1} \exp\left(-\frac{b_0}{\boldsymbol{\Gamma}_{c,ij}}\right), \tag{47}$$

$$p(\boldsymbol{\Gamma}_d) = \prod_{i=1}^{m} \prod_{j=1}^{p} \frac{a_0^{b_0}}{\Gamma(a_0)} \boldsymbol{\Gamma}_{d,ij}^{-a_0-1} \exp\left(-\frac{b_0}{\boldsymbol{\Gamma}_{d,ij}}\right). \tag{48}$$

## B  DETAILED MATHEMATICAL DERIVATION

### B.1  DERIVATION OF EQUATION 16

Given the conditional independence between the variables, we can derive

$$H(\boldsymbol{\Theta} \mid \boldsymbol{\Theta}^k)$$
$$= \mathbb{E}_{\boldsymbol{X} \sim p(\boldsymbol{X}|\boldsymbol{Y},\boldsymbol{\Theta}^k)}[\log p(\boldsymbol{Y}, \boldsymbol{X} \mid \boldsymbol{\Theta})p(\boldsymbol{\Theta})]$$
$$= \mathbb{E}_{\boldsymbol{X} \sim p(\boldsymbol{X}|\boldsymbol{Y},\boldsymbol{\Theta}^k)}[\log p(\boldsymbol{Y} \mid \boldsymbol{X}, \boldsymbol{\Theta})p(\boldsymbol{X} \mid \boldsymbol{\Theta})p(\boldsymbol{\Theta})]$$
$$= \mathbb{E}_{\boldsymbol{X} \sim p(\boldsymbol{X}|\boldsymbol{Y},\boldsymbol{\Theta}^k)}[\log p(\boldsymbol{Y} \mid \boldsymbol{X}, \boldsymbol{C}, \boldsymbol{D}, \boldsymbol{Q})p(\boldsymbol{X} \mid \boldsymbol{A}, \boldsymbol{B}, \boldsymbol{R})p(\boldsymbol{A}, \boldsymbol{\Gamma}_a)p(\boldsymbol{B}, \boldsymbol{\Gamma}_b)p(\boldsymbol{C}, \boldsymbol{\Gamma}_c)p(\boldsymbol{D}, \boldsymbol{\Gamma}_d)]$$
$$= \underbrace{\mathbb{E}_{\boldsymbol{X} \sim p(\boldsymbol{X}|\boldsymbol{Y},\boldsymbol{\Theta}^k)}[\log p(\boldsymbol{X} \mid \boldsymbol{A}, \boldsymbol{B}, \boldsymbol{R})]}_{H_1(\boldsymbol{A},\boldsymbol{B},\boldsymbol{R})} + \underbrace{\mathbb{E}_{\boldsymbol{X} \sim p(\boldsymbol{X}|\boldsymbol{Y},\boldsymbol{\Theta}^k)}[\log p(\boldsymbol{Y} \mid \boldsymbol{X}, \boldsymbol{C}, \boldsymbol{D}, \boldsymbol{Q})]}_{H_2(\boldsymbol{C},\boldsymbol{D},\boldsymbol{Q})}$$
$$+ \underbrace{\mathbb{E}_{\boldsymbol{X} \sim p(\boldsymbol{X}|\boldsymbol{Y},\boldsymbol{\Theta}^k)}[\log p(\boldsymbol{A}, \boldsymbol{\Gamma}_a)p(\boldsymbol{B}, \boldsymbol{\Gamma}_b)p(\boldsymbol{C}, \boldsymbol{\Gamma}_c)p(\boldsymbol{D}, \boldsymbol{\Gamma}_d)]}_{H_3(\boldsymbol{A},\boldsymbol{B},\boldsymbol{C},\boldsymbol{D},\boldsymbol{\Gamma}_a,\boldsymbol{\Gamma}_b,\boldsymbol{\Gamma}_c,\boldsymbol{\Gamma}_d)}. \tag{49}$$

**Explicit mathematical expression of $H_1(\boldsymbol{A}, \boldsymbol{B}, \boldsymbol{R})$.** Based on equation 1 and the chain rule in probability, we can derive

$$p(\boldsymbol{X} \mid \boldsymbol{A}, \boldsymbol{B}, \boldsymbol{R})$$
$$= p(\boldsymbol{x}_0) \prod_{t=1}^{T} p(\boldsymbol{x}_t \mid \boldsymbol{x}_{t-1}, \boldsymbol{A}, \boldsymbol{B}, \boldsymbol{R}) \tag{50}$$
$$\propto \prod_{t=1}^{T} \mid \boldsymbol{R} \mid^{-\frac{1}{2}} \exp\left(-\frac{(\boldsymbol{x}_t - \boldsymbol{A}\boldsymbol{x}_{t-1} - \boldsymbol{B}\boldsymbol{u}_t)'\boldsymbol{R}^{-1}(\boldsymbol{x}_t - \boldsymbol{A}\boldsymbol{x}_{t-1} - \boldsymbol{B}\boldsymbol{u}_t)}{2}\right). \tag{51}$$

Hence,

$$
\begin{aligned}
&H_1(\boldsymbol{A}, \boldsymbol{B}, \boldsymbol{R}) \\
&= \mathbb{E}_{\boldsymbol{X} \sim p(\boldsymbol{X}|\boldsymbol{Y}, \boldsymbol{\Theta}^k)}[\log p(\boldsymbol{X} \mid \boldsymbol{A}, \boldsymbol{B}, \boldsymbol{R})] \\
&= \mathbb{E}_{\boldsymbol{X} \sim p(\boldsymbol{X}|\boldsymbol{Y}, \boldsymbol{\Theta}^k)}\left[ -\frac{T \log \mid \boldsymbol{R} \mid + \sum_{t=1}^{T}(\boldsymbol{x}_t - \boldsymbol{A}\boldsymbol{x}_{t-1} - \boldsymbol{B}\boldsymbol{u}_t)'\boldsymbol{R}^{-1}(\boldsymbol{x}_t - \boldsymbol{A}\boldsymbol{x}_{t-1} - \boldsymbol{B}\boldsymbol{u}_t)}{2} \right] \\
&= -\frac{T \log \mid \boldsymbol{R} \mid + \sum_{t=1}^{T}\mathbb{E}_{\boldsymbol{X} \sim p(\boldsymbol{X}|\boldsymbol{Y}, \boldsymbol{\Theta}^k)}(\boldsymbol{x}_t - \boldsymbol{A}\boldsymbol{x}_{t-1} - \boldsymbol{B}\boldsymbol{u}_t)'\boldsymbol{R}^{-1}(\boldsymbol{x}_t - \boldsymbol{A}\boldsymbol{x}_{t-1} - \boldsymbol{B}\boldsymbol{u}_t)}{2} \\
&= -\frac{T \log \mid \boldsymbol{R} \mid + \sum_{t=1}^{T}\left(\boldsymbol{m}_t^k - \boldsymbol{A}\boldsymbol{m}_{t-1}^k - \boldsymbol{B}\boldsymbol{u}_t\right)'\boldsymbol{R}^{-1}\left(\boldsymbol{m}_t^k - \boldsymbol{A}\boldsymbol{m}_{t-1}^k - \boldsymbol{B}\boldsymbol{u}_t\right)}{2} \\
&\quad - \frac{\sum_{t=1}^{T}\left(\mathrm{Tr}(\boldsymbol{R}^{-1}\boldsymbol{P}_t^k) - \mathrm{Tr}(\boldsymbol{R}^{-1}\boldsymbol{A}\boldsymbol{P}_{t,t-1}^k) - \mathrm{Tr}(\boldsymbol{A}'\boldsymbol{R}^{-1}\boldsymbol{P}_{t,t-1}^k) + \mathrm{Tr}(\boldsymbol{A}'\boldsymbol{R}^{-1}\boldsymbol{A}\boldsymbol{P}_{t-1}^k)\right)}{2}. \quad (52)
\end{aligned}
$$

**Explicit mathematical expression of** $H_2(\boldsymbol{C}, \boldsymbol{D}, \boldsymbol{Q})$**.** Based on equation 2, we can derive

$$
\begin{aligned}
&p(\boldsymbol{Y} \mid \boldsymbol{X}, \boldsymbol{C}, \boldsymbol{D}, \boldsymbol{Q}) \\
&= \prod_{t=1}^{T} p(\boldsymbol{y}_t \mid \boldsymbol{x}_t, \boldsymbol{C}, \boldsymbol{D}) \\
&\propto \prod_{t=1}^{T} \mid \boldsymbol{Q} \mid^{-\frac{1}{2}} \exp\left( -\frac{(\boldsymbol{y}_t - \boldsymbol{C}\boldsymbol{x}_t - \boldsymbol{D}\boldsymbol{u}_t)'\boldsymbol{Q}^{-1}(\boldsymbol{y}_t - \boldsymbol{C}\boldsymbol{x}_t - \boldsymbol{D}\boldsymbol{u}_t)}{2} \right). \quad (53)
\end{aligned}
$$

Hence,

$$
\begin{aligned}
&H_2(\boldsymbol{C}, \boldsymbol{D}, \boldsymbol{Q}) \\
&= \mathbb{E}_{\boldsymbol{X} \sim p(\boldsymbol{X}|\boldsymbol{Y}, \boldsymbol{\Theta}^k)}[\log p(\boldsymbol{Y} \mid \boldsymbol{X}, \boldsymbol{C}, \boldsymbol{D}, \boldsymbol{Q})] \\
&= \mathbb{E}_{\boldsymbol{X} \sim p(\boldsymbol{X}|\boldsymbol{Y}, \boldsymbol{\Theta}^k)}\left[ -\frac{T \log \mid \boldsymbol{Q} \mid + \sum_{t=1}^{T}(\boldsymbol{y}_t - \boldsymbol{C}\boldsymbol{x}_t - \boldsymbol{D}\boldsymbol{u}_t)'\boldsymbol{Q}^{-1}(\boldsymbol{y}_t - \boldsymbol{C}\boldsymbol{x}_t - \boldsymbol{D}\boldsymbol{u}_t)}{2} \right] \\
&= -\frac{T \log \mid \boldsymbol{Q} \mid + \sum_{t=1}^{T}\mathbb{E}_{\boldsymbol{X} \sim p(\boldsymbol{X}|\boldsymbol{Y}, \boldsymbol{\Theta}^k)}(\boldsymbol{y}_t - \boldsymbol{C}\boldsymbol{x}_t - \boldsymbol{D}\boldsymbol{u}_t)'\boldsymbol{Q}^{-1}(\boldsymbol{y}_t - \boldsymbol{C}\boldsymbol{x}_t - \boldsymbol{D}\boldsymbol{u}_t)}{2} \\
&= -\frac{T \log \mid \boldsymbol{Q} \mid + \sum_{t=1}^{T}\left((\boldsymbol{y}_t - \boldsymbol{C}\boldsymbol{m}_t^k - \boldsymbol{D}\boldsymbol{u}_t)'\boldsymbol{Q}^{-1}(\boldsymbol{y}_t - \boldsymbol{C}\boldsymbol{m}_t^k - \boldsymbol{D}\boldsymbol{u}_t) + \mathrm{Tr}(\boldsymbol{C}'\boldsymbol{Q}^{-1}\boldsymbol{C}\boldsymbol{P}_t^k)\right)}{2}. \\
&\quad\quad (54)
\end{aligned}
$$

**Explicit mathematical expression of** $H_3(\boldsymbol{A}, \boldsymbol{B}, \boldsymbol{C}, \boldsymbol{D}, \boldsymbol{\Gamma}_a, \boldsymbol{\Gamma}_b, \boldsymbol{\Gamma}_c, \boldsymbol{\Gamma}_d)$**.** Based on the priors imposed on the system matrices and corresponding hyperparameters, we can derive:

$$
\begin{aligned}
&p(\boldsymbol{A}, \boldsymbol{\Gamma}_a)p(\boldsymbol{B}, \boldsymbol{\Gamma}_b)p(\boldsymbol{C}, \boldsymbol{\Gamma}_c)p(\boldsymbol{D}, \boldsymbol{\Gamma}_d) \\
&= p(\boldsymbol{A} \mid \boldsymbol{\Gamma}_a)p(\boldsymbol{\Gamma}_a)p(\boldsymbol{B} \mid \boldsymbol{\Gamma}_b)p(\boldsymbol{\Gamma}_b)p(\boldsymbol{C} \mid \boldsymbol{\Gamma}_c)p(\boldsymbol{\Gamma}_c)p(\boldsymbol{D} \mid \boldsymbol{\Gamma}_d)p(\boldsymbol{\Gamma}_d) \\
&\propto \prod_{i=1}^{n}\prod_{j=1}^{n} \boldsymbol{\Gamma}_{a,ij}^{-\frac{2a_0+3}{2}} \exp\left( -\frac{\boldsymbol{A}_{ij}^2 + 2b_0}{2\boldsymbol{\Gamma}_{a,ij}} \right) \times \prod_{i=1}^{n}\prod_{j=1}^{p} \boldsymbol{\Gamma}_{b,ij}^{-\frac{2a_0+3}{2}} \exp\left( -\frac{\boldsymbol{B}_{ij}^2 + 2b_0}{2\boldsymbol{\Gamma}_{b,ij}} \right) \\
&\quad \times \prod_{i=1}^{m}\prod_{j=1}^{n} \boldsymbol{\Gamma}_{c,ij}^{-\frac{2a_0+3}{2}} \exp\left( -\frac{\boldsymbol{C}_{ij}^2 + 2b_0}{2\boldsymbol{\Gamma}_{c,ij}} \right) \times \prod_{i=1}^{m}\prod_{j=1}^{p} \boldsymbol{\Gamma}_{d,ij}^{-\frac{2a_0+3}{2}} \exp\left( -\frac{\boldsymbol{D}_{ij}^2 + 2b_0}{2\boldsymbol{\Gamma}_{d,ij}} \right). \quad (55)
\end{aligned}
$$

Hence, we have

$$H_3(\boldsymbol{A}, \boldsymbol{B}, \boldsymbol{C}, \boldsymbol{D}, \boldsymbol{\Gamma}_a, \boldsymbol{\Gamma}_b, \boldsymbol{\Gamma}_c, \boldsymbol{\Gamma}_d)$$
$$= \mathbb{E}_{\boldsymbol{X} \sim p(\boldsymbol{X}|\boldsymbol{Y}, \boldsymbol{\Theta}^k)}[\log p(\boldsymbol{A}, \boldsymbol{\Gamma}_a) p(\boldsymbol{B}, \boldsymbol{\Gamma}_b) p(\boldsymbol{C}, \boldsymbol{\Gamma}_c) p(\boldsymbol{D}, \boldsymbol{\Gamma}_d)].$$

$$= -\sum_{i=1}^{n} \sum_{j=1}^{n} \left( \frac{(2a_0 + 3) \log |\boldsymbol{\Gamma}_{a,ij}|}{2} + \frac{\boldsymbol{A}_{ij}^2 + 2b_0}{2\boldsymbol{\Gamma}_{a,ij}} \right)$$

$$- \sum_{i=1}^{n} \sum_{j=1}^{p} \left( \frac{(2a_0 + 3) \log |\boldsymbol{\Gamma}_{b,ij}|}{2} + \frac{\boldsymbol{B}_{ij}^2 + 2b_0}{2\boldsymbol{\Gamma}_{b,ij}} \right)$$

$$- \sum_{i=1}^{m} \sum_{j=1}^{n} \left( \frac{(2a_0 + 3) \log |\boldsymbol{\Gamma}_{c,ij}|}{2} + \frac{\boldsymbol{C}_{ij}^2 + 2b_0}{2\boldsymbol{\Gamma}_{c,ij}} \right)$$

$$- \sum_{i=1}^{m} \sum_{j=1}^{p} \left( \frac{(2a_0 + 3) \log |\boldsymbol{\Gamma}_{d,ij}|}{2} + \frac{\boldsymbol{D}_{ij}^2 + 2b_0}{2\boldsymbol{\Gamma}_{d,ij}} \right). \tag{56}$$

## B.2 DERIVATION OF EQUATION 22

To provide an efficient closed-form update rule for $\boldsymbol{A}$, we only keep the diagonal components of $\boldsymbol{R}^k$ and set the others to zero during the optimization process. As such, we have

$$H_1(\boldsymbol{A}, \boldsymbol{B}^k, \boldsymbol{R}^k) + H_3(\boldsymbol{A}, \boldsymbol{B}^k, \boldsymbol{C}^k, \boldsymbol{D}^k, \boldsymbol{\Gamma}_a^k, \boldsymbol{\Gamma}_b^k, \boldsymbol{\Gamma}_c^k, \boldsymbol{\Gamma}_d^k)$$

$$= -\frac{\sum_{t=1}^{T} \sum_{r=1}^{n} \left( \boldsymbol{R}_{rr}^k \right)^{-1} \left( \boldsymbol{m}_{t,r}^k - \boldsymbol{A}_r \boldsymbol{m}_{t-1}^k - \boldsymbol{B}_r^k \boldsymbol{u}_t \right)^2}{2} - \sum_{r=1}^{n} \frac{\boldsymbol{A}_r \overline{\boldsymbol{\Gamma}}_{a,r}^{kd} \boldsymbol{A}_r'}{2}$$

$$- \frac{\sum_{t=1}^{T} \sum_{r=1}^{n} \left( \text{Tr} \left( \boldsymbol{A}_r' \left( \boldsymbol{R}_{rr}^k \right)^{-1} \boldsymbol{A}_r \boldsymbol{P}_{t-1}^k \right) - 2 \left( \boldsymbol{R}_{rr}^k \right)^{-1} \boldsymbol{A}_r \left( \boldsymbol{P}_{t,t-1,r}^k \right)' \right)}{2} + c, \tag{57}$$

where $c$ is the term unrelated to $\boldsymbol{A}$. Hence, we can calculate the derivative of $H(\boldsymbol{\Theta} \mid \boldsymbol{\Theta}^k)$ with respect to $\boldsymbol{A}_r$ at the $k$th iteration as follows:

$$\frac{\partial H_1(\boldsymbol{A}, \boldsymbol{B}^k, \boldsymbol{R}^k)}{\partial \boldsymbol{A}_r} + \frac{\partial H_3(\boldsymbol{A}, \boldsymbol{B}^k, \boldsymbol{C}^k, \boldsymbol{D}^k, \boldsymbol{\Gamma}_a^k, \boldsymbol{\Gamma}_b^k, \boldsymbol{\Gamma}_c^k, \boldsymbol{\Gamma}_d^k)}{\partial \boldsymbol{A}_r}$$

$$= \sum_{t=1}^{T} \left( \boldsymbol{R}_{rr}^k \right)^{-1} \left( \boldsymbol{m}_{t,r}^k - \boldsymbol{A}_r \boldsymbol{m}_{t-1}^k - \boldsymbol{B}_r^k \boldsymbol{u}_t \right) \left( \boldsymbol{m}_{t-1}^k \right)' - \boldsymbol{A}_r \overline{\boldsymbol{\Gamma}}_{a,r}^{kd}$$

$$- \sum_{t=1}^{T} \left( \boldsymbol{R}_{rr}^k \right)^{-1} \left( \boldsymbol{A}_r \boldsymbol{P}_{t-1}^k - \boldsymbol{P}_{t,t-1,r}^k \right)$$

$$= \sum_{t=1}^{T} \left( \boldsymbol{R}_{rr}^k \right)^{-1} \left( \boldsymbol{P}_{t,t-1,r}^k + (\boldsymbol{m}_{t,r}^k - \boldsymbol{A}_r \boldsymbol{m}_{t-1}^k - \boldsymbol{B}_r^k \boldsymbol{u}_t) \left( \boldsymbol{m}_{t-1}^k \right)' - \boldsymbol{A}_r \boldsymbol{P}_{t-1}^k \right) - \boldsymbol{A}_r \overline{\boldsymbol{\Gamma}}_{a,r}^{kd}. \tag{58}$$

Setting equation 58 to zero leads to

$$\sum_{t=1}^{T} \left( \boldsymbol{R}_{rr}^k \right)^{-1} \left( \boldsymbol{A}_r \boldsymbol{m}_{t-1}^k \left( \boldsymbol{m}_{t-1}^k \right)' + \boldsymbol{A}_r \boldsymbol{P}_{t-1}^k \right) + \boldsymbol{A}_r \overline{\boldsymbol{\Gamma}}_{a,r}^{kd}$$

$$= \sum_{t=1}^{T} \left( \boldsymbol{R}_{rr}^k \right)^{-1} \left( \boldsymbol{P}_{t,t-1,r}^k + (\boldsymbol{m}_{t,r}^k - \boldsymbol{B}_r^k \boldsymbol{u}_t) \left( \boldsymbol{m}_{t-1}^k \right)' \right), \tag{59}$$

Hence, we can update $\boldsymbol{A}$ at the $k$th iteration as follows:

$$\boldsymbol{A}_r^{k+1} = \left( \sum_{t=1}^{T} \left( (\boldsymbol{m}_{t,r}^k - \boldsymbol{B}_r^k \boldsymbol{u}_t)(\boldsymbol{m}_{t-1}^k)' + \boldsymbol{P}_{t,t-1,r}^k \right) \right)$$

$$\times \left( \sum_{t=1}^{T} \left( \boldsymbol{P}_{t-1}^k + \boldsymbol{m}_{t-1}^k \left( \boldsymbol{m}_{t-1}^k \right)' \right) + \boldsymbol{R}_{rr}^k \boldsymbol{\Gamma}_{a,r}^{kd} \right)^{-1}. \tag{60}$$

## C EQUIVALENT REALIZATION OF LDSS

Based on the transformed coordinates, we can derive

$$\overline{\boldsymbol{x}}_t = \boldsymbol{P}\boldsymbol{x}_t = \boldsymbol{P}\boldsymbol{A}\boldsymbol{x}_{t-1} + \boldsymbol{P}\boldsymbol{B}\boldsymbol{u}_t + \boldsymbol{P}\boldsymbol{\varepsilon}_t = \left(\boldsymbol{P}\boldsymbol{A}\boldsymbol{P}^{-1}\right)\overline{\boldsymbol{x}}_{t-1} + \left(\boldsymbol{P}\boldsymbol{B}\right)\boldsymbol{u}_t + \boldsymbol{P}\boldsymbol{\varepsilon}_t, \quad (61)$$

$$\boldsymbol{y}_t = \boldsymbol{C}\boldsymbol{x}_t + \boldsymbol{D}\boldsymbol{u}_t + \boldsymbol{\omega}_t = \left(\boldsymbol{C}\boldsymbol{P}^{-1}\right)\overline{\boldsymbol{x}}_t + \boldsymbol{D}\boldsymbol{u}_t + \boldsymbol{\omega}_t. \quad (62)$$

Hence, an equivalent realization of the original LDSs is as follows:

$$\overline{\boldsymbol{x}}_t = \overline{\boldsymbol{A}}\overline{\boldsymbol{x}}_{t-1} + \overline{\boldsymbol{B}}\boldsymbol{u}_t + \overline{\boldsymbol{\varepsilon}}_t, \quad (63)$$

$$\boldsymbol{y}_t = \overline{\boldsymbol{C}}\overline{\boldsymbol{x}}_t + \boldsymbol{D}\boldsymbol{u}_t + \boldsymbol{\omega}_t, \quad (64)$$

where $\overline{\boldsymbol{A}} = \boldsymbol{P}\boldsymbol{A}\boldsymbol{P}^{-1}, \overline{\boldsymbol{B}} = \boldsymbol{P}\boldsymbol{B}, \overline{\boldsymbol{C}} = \boldsymbol{C}\boldsymbol{P}^{-1}$, and $\overline{\boldsymbol{\varepsilon}}_t = \boldsymbol{P}\boldsymbol{\varepsilon}_t$.

