# OpenReview forum: "Learning Linear Dynamical Systems with Sparse System Matrices"
_ICLR.cc/2025/Conference — Submitted to ICLR 2025_

### Official Review · Reviewer_jwHG · 2024-10-31

**Soundness:** 3
**Presentation:** 2
**Contribution:** 2
**Rating:** 6
**Confidence:** 3

**Summary:**

The paper addresses the learning of linear dynamical systems (LDSs) with sparse system matrices. The authors propose an expectation–maximization (EM) algorithm that incorporates sparsity-promoting priors on system matrices, enabling a maximum a posteriori (MAP) estimation of both hidden states and system matrices from noisy observations. The authors validate their algorithm with experiments on synthetic and real-world datasets, demonstrating better performance over traditional methods such as PEM, 4SID, and MLE

**Strengths:**

1. The paper is well-motivated, with a thorough analysis of related work that situates the proposed method within the context of recent advances in LDS learning.
2. The mathematical derivations are sound, providing a solid theoretical foundation for the proposed approach.

**Weaknesses:**

1. The definition of matrix sparsity could be clarified. While the authors introduce a sparsity-promoting prior, it would be beneficial to show examples of what matrices look like after imposing the Student’s t-distribution to better illustrate its effect.
2. The applicability of this method in higher-dimensional cases is unclear. High-dimensional systems often exhibit sparse structures, and it would be insightful to explore how the method scales or adapts to such scenarios.
3. The paper is mathematically dense, which could challenge readers unfamiliar with LDSs. An additional table summarizing the notations may enhance readability and aid understanding.
4. The synthetic experiment in Section 5.1 appears overly simplified, potentially limiting insights into the method’s performance. It would be helpful to see additional experiments with more complex or higher-dimensional system matrices.
5. For the real-world dataset experiments, it would be valuable to include datasets where the true underlying system matrices are known. This would allow for a direct comparison between the learned and true matrices or at least an evaluation of the topological structure similarity.

**Questions:**

1. Could the authors provide a brief explanation of the Occam’s Razor principle?
2. While the proposed method aims to learn the system matrices, even in the simple synthetic example in Section 5.1, the learned matrices (B, C, R, Q) differ from the true ones. If the primary goal is to predict observations, are there more direct approaches that could yield higher prediction accuracy?

---

> ### Author Response · Authors · 2024-11-21
> **Rebuttal #1**
>
> We thank the reviewer for taking the time to review our paper. We are glad that you think the paper is well-motivated and the mathematical derivations are sound. We address your concerns and questions below.
>
> **Response to Weaknesses**
>
> >**The definition of matrix sparsity could be clarified. While the authors introduce a sparsity-promoting prior, it would be beneficial to show examples of what matrices look like after imposing the Student’s t-distribution to better illustrate its effect.**
>
> * We appreciate your suggestion to clarify the definition of matrix sparsity.  In our work, matrix sparsity refers to the proportion of zero elements in the matrix. Basically, a matrix with sparsity less than 1 can be considered a sparse matrix. Based on your comment, we will clarify the definition of matrix sparsity in the revised version.
> * In fact, Section 5.1 has provided such an example to show the learned system matrices of classical algorithms and the learned system matrices of the proposed algorithm on a two-dimensional system. Without considering the sparsity constraint, the learned system matrices of classical algorithms differ significantly from the true ones due to the similarity transformation. By introducing the sparsity-promoting prior, the proposed algorithm learns LDSs by balancing model complexity and modeling error. As such, the nonsingular matrix in the similarity transformation  is typically restricted to be a generalized permutation matrix to avoid increasing model complexity, as this matrix only scales the values of the system matrices or permutes their rows or columns. As a result,  the learned system matrices of the proposed algorithm  preserve the topological structure of the true ones.  It is worth noting that although they differ in terms of numerical values, this discrepancy is solely a result of the scaled definition of the system states, rather than a failure to capture the underlying system dynamics.
>
> > **The applicability of this method in higher-dimensional cases is unclear. High-dimensional systems often exhibit sparse structures, and it would be insightful to explore how the method scales or adapts to such scenarios.**
>
> * Based on your constructive comment, we further test all the algorithms on a 15-dimensional system. As a result, each system matrix has 225 unknown parameters. Specifically, we consider $A$ to be an anti-diagonal matrix with nonzero components equal to 0.9 and set $B=C=D=1.5I_{15}$ and $R=Q\sim \mathcal N(0,0.81I_{15})$. Because these system matrices are extremely sparse,  accurately learning their topological structures is quite challenging. As for algorithm implementation, we collect 2400 data points and initialize $A, B, C, D, R,$ and $Q$ as identity matrices.  The learned parameters less than the
> threshold 0.002 are removed from the result.  The following table records whether the learned system matrices  preserve the topological structure of the true ones and the mean relative error (MRE) defined in the paper:
> | Method | Ours |  PEM | 4SID | MLE |
> |----------|----------|----------|----------|----------|
> | Topology Preservation |  $\checkmark$ | $\times$ | $\times$ | $\times$ |
> | MRE |  **13.66\%**|  19.96\% | 25.99\% | 13.91\%|
>
>     Obviously, the experimental results on higher-dimensional systems are consistent with those observed in low-dimensional systems. The  proposed algorithm can preserve the inherent topological structure  among the variables and achieves the lowest MRE.
>
> > **The paper is mathematically dense, which could challenge readers unfamiliar with LDSs. An additional table summarizing the notations may enhance readability and aid understanding.**
>
> * We agree with the reviewer that an additional table summarizing the notations on LDSs can enhance readability. Due to the limited space, we will give the table in Appendix in the revised version as follows:
> | Notation     | Description   |
> |:------------|:------------|
> | $u_t$       | Input at time $t$     |
> | $x_t$      | System state  at time $t$     |
> | $y_t$       | Output  at time $t$     |
> | $\varepsilon_t$       | Process noise at time $t$     |
> | $\omega_t$       | Measurement noise at time $t$     |
> | $A,B,C,D$       | System matrices   |
> | $\Gamma_a,\Gamma_b,\Gamma_c,\Gamma_d$       | Hyperparameter matrices   |
> | $R$     | The covariance of process noise     |
> | $Q$     | The covariance of measurement noise     |
> | $m_t$     | The mean of $x_t$   |
> | $P_t$     | The covariance of $x_t$   |
> | $P_{t,t-1}$     | The covariance between $x_t$  and $x_{t-1}$  |
> | $\Theta$     | The set consisting of $A,B,C,D,R,Q,\Gamma_a,\Gamma_b,\Gamma_c,\Gamma_d$   |
> | $X$     | The matrix with the $i$-th column being $x_i$   |
> | $Y$     | The matrix with the $i$-th column being $y_i$   |
> | $p(Y\mid \Theta)$     | Likelihood function  of data |
> | $p(\Theta)$     | Prior distribution of $\Theta$  |
> | $p(\Theta\mid Y)$     | Posterior distribution of   $\Theta$ |

---

> ### Author Response · Authors · 2024-11-21
> **Rebuttal #2**
>
> > **The synthetic experiment in Section 5.1 appears overly simplified, potentially limiting insights into the method’s performance. It would be helpful to see additional experiments with more complex or higher-dimensional system matrices.**
>
> * See above
>
> > **For the real-world dataset experiments, it would be valuable to include datasets where the true underlying system matrices are known. This would allow for a direct comparison between the learned and true matrices or at least an evaluation of the topological structure similarity.**
>
> * We fully agree with the reviewer's point. However,  to the best of our knowledge,  there are currently no available databases or published papers that provide such a dataset. We are actively looking for potential sources, but at this time, we are unable to conduct experiments with the desired dataset. Nevertheless, we believe that the experimental results on both simulated and real-world datasets presented in the paper provide strong evidence that the proposed algorithm outperforms classical ones on learning LDSs with sparse system matrices.
>
> **Response to Questions**
>
> > **Could the authors provide a brief explanation of the Occam’s Razor principle?**
>
> * Basically, the _Occam’s Razor_ principle states that the simplest explanation is usually the best when faced with multiple competing explanations. Hence, it recommends searching for explanations constructed with the smallest possible set of elements. In the context of system identification, this principle encourages that  the learned LDSs should include the minimally required parameters to explain input-output data.
>
> > **While the proposed method aims to learn the system matrices, even in the simple synthetic example in Section 5.1, the learned matrices (B, C, R, Q) differ from the true ones. If the primary goal is to predict observations, are there more direct approaches that could yield higher prediction accuracy?**
>
> * As discussed in Section 4, there are multiple equivalent realizations of the original LDSs due to the similarity transformation. Hence, to the best of our knowledge, existing methods cannot exactly learn system matrices of LDSs. By introducing sparsity-promoting  priors, our proposed method  learns LDSs by balancing model complexity and modeling error. Hence, the nonsingular matrix of the similar transformation is typically restricted to be a generalized permutation matrix to avoid increasing model complexity. The learned system matrices of the proposed algorithm can thus preserve the topological structure of the true ones compared to the classical algorithms. While they differ in terms of numerical values, this discrepancy is solely due to the scaled definition of the hidden system states, rather than a failure to capture the underlying system dynamics. As a result,  our approach is particularly valuable for revealing the system's internal structure and interaction rules, which may not be the primary focus of purely predictive algorithms. In addition, note that we actually use prediction accuracy to evaluate the performance of the learned system matrices due to the lack of ground truth, rather than focusing on predicting observations.

---

> > ### Comment · Reviewer_jwHG · 2024-12-02
> > **Thank you for your response**
> >
> > Thank you for answering my questions. While the authors stated that their work aims to reveal the system's internal structure and interaction rules, the primary metric used is prediction accuracy, which does not convincingly validate this main claim. For the added higher-dimensional cases, the only provided result is MRE, which does not sufficiently demonstrate the learned topological structures. Therefore, I will maintain my initial rating.

---

> > > ### Author Response · Authors · 2024-12-02
> > > **Rebuttal #3**
> > >
> > > We appreciate the reviewer for the prompt feedback.
> > >
> > > * For the added higher-dimensional cases, we **not only** report the MRE **but also** demonstrate whether the learned system matrices preserve the topological structure of the true ones (i.e., **Topology Preservation**). As shown in the table, only the learned system matrices from the proposed algorithm are able to preserve the topological structure of the true matrices.
> > >
> > > * For the real-world datasets,  we cannot compare the learned system matrices with the true ones due to the lack of ground truth. Hence, we use the prediction accuracy as the metric.

---

> > > > ### Comment · Reviewer_jwHG · 2024-12-02
> > > > **Response to authors**
> > > >
> > > > Thank you for providing further clarification. As my concerns have been addressed, I am increasing my rating.

---

> > > > > ### Author Response · Authors · 2024-12-02
> > > > >
> > > > > We really appreciate the reviewer for increasing the score. We  also appreciate your time and thoughtful comments throughout the review process.

---

### Official Review · Reviewer_Kt5E · 2024-10-31

**Soundness:** 3
**Presentation:** 3
**Contribution:** 2
**Rating:** 5
**Confidence:** 4

**Summary:**

The article addresses the problem of identifying system matrices $A,B,C,D$  and noise covariances $R$ and $Q$ assumed to be normal for the dynamical system:

$$\begin{array}{rll}
x_t&=&Ax_{t-1}+Bu_t+\epsilon_t\\
y_t&=& Cx_t+Du_t+\omega_t
\end{array}$$

from the data $\{(u_t,y_t)\}$ over a time-horizon $T.$

To identify the matrices $A,B.C,D$ the authors propose to use priors that promote sparsity; the parameters, dependent on $\Gamma_a, \Gamma_b,\Gamma_c$ and $\Gamma_d$  of the priors are also learnt in the apporach presented.

In the approach, the priors are initialized to be noninformative. The approach is as follows:

(i) The first step of the iteration: if the set of parameters governing the model given by $\Theta^{(k)}=(A,B,C,D,R,Q,\Gamma_a, \Gamma_b,\Gamma_c$ and $\Gamma_d)$ is provided then one can determine the distribution of the state $x_t$ from measurements $y_t$ using a standard kalman filter like update.  Dynamics are linear and noise sources gaussian and thus analytical expression for the conditional density of $x_t$ can be obtained.

(ii) In the second step of the iteration, the parameters in $\Theta$ are updated by maximizing a posterior function of the  $\Theta$, $H(\Theta|\Theta^{(k)})$  wherein the conidtional distribution, $p(x_t|y,\Theta^{(k)})$ is leveraged to maximize

$$H(\Theta|\Theta^{(k)})=\mathbb{E}[\log~p(Y,X|\Theta)p(\Theta)]$$

with the expection taken over the pdf obtained in step (i).  The $H$ can be deteremined  as analytical expressions in terms of $(A,B,C,D,R,Q,\Gamma_a, \Gamma_b,\Gamma_c,\Gamma_d)$. The authors proceeed to set the derivatives of $H$ with respect to the parameters to zero. Here, the resulting equations do not admit an explicit solutions; the authors make assumption on the noise covariances being diagonal to reach an explicit closed-form solutions for derivatives with respect to the $A,B,C,D$ matrices; while taking deribatives with respect to $R$, the symmetry of the matrix $R$ is incorporated and respected.

The simulation section demonstrates that the inclusion of the priors promotes preserving structure of the system matrices.

**Strengths:**

The idea of imposing sparsity promoting priors to preserve structure of the system matrices for linear dynamical systems is interesting and possibly impactful.

**Weaknesses:**

(1) The methodology more or less mirrors the development in  "Sparse Bayesian Learning and the Relevance Vector Machine" article by Tipping. The novelty seems incremental

(2) Another of authors article "An Iterative Min-Min Optimization Method for Sparse Bayesian Learning", seems to be very related to the present article; specifically it does provide insights into why the priors on the weights promotes sparsity by comparing with regularizers employed. Given the earlier works, the contribution of the present article becomes more incremental

**Questions:**

(1) Can the authors delineate the novelty with a comparison with the ideas presented in "Sparse Bayesian Learning and the Relevance Vector Machine" article by Tipping?

(2) Can the authors state which aspects of this article differ in a substantitive manner over analysis in "An Iterative Min-Min Optimization Method for Sparse Bayesian Learning"? Which new insights are found?

(3) Can the authors indicate why the priors promote sturcture of the matrices in contrast to sparsity alone?

---

> ### Author Response · Authors · 2024-11-21
> **Rebuttal #1**
>
> We thank the reviewer for taking the time to review our paper. We are glad that you think the idea of imposing sparsity promoting priors to preserve structure of the system matrices for linear dynamical systems (LDSs) is interesting and possibly impactful. We address your questions and concerns below.
>
> **Response to Weaknesses**
> > **The methodology more or less mirrors the development in "Sparse Bayesian Learning and the Relevance Vector Machine" article by Tipping. The novelty seems incremental.**
> * Thanks for your thoughtful feedback and we appreciate the opportunity to clarify the differences between our research and the work by Tipping. While both studies use sparsity-promoting priors to learn systems with sparse topology,  the models and problems addressed are fundamentally different. Specifically:
>     1. Tipping considers the the problem of learning $y_t=\Phi(u_t)w+\varepsilon_t$ from  data $(u_t,y_t)_{t=1}^T$, where $y_t$ is the output,  $\Phi(u_t)$ is the library consisting of  basis functions,  $\varepsilon_t$ is the measurement noise, and $w$ is the unknown weight vector.
>     2. In this paper,  we consider the problem of learning state-space representations of linear dynamical systems (LDSs) from data $(u_t, y_t)^T_{t=1}$, which consists of the state transition equation $x_{t}=Ax_{t-1}+Bu_t+\varepsilon_t$ and the observation equation $y_{t}=Cx_{t}+Du_t+\omega_t$, where $x_t$ is the system state, $u_t$ is the input, $y_t$ is the output, $\varepsilon_t$ and $\omega_t$ are the process and measurement noise, respectively, and $A,B,C,$ and $D$ are the unknown system matrices.
>
>     Compared to Tipping's work, learning LDSs with sparse system matrices involves a significant challenge: the system state $x_t$, which links the state transition equation and observation equation, is unknown. Hence, the methodology in our paper actually addresses the problem of learning sparse system matrices and estimating hidden system states simultaneously. The novelty of our method lies in leveraging the expectation-maximization framework to seamlessly integrate the strengths of the Rauch–Tung–Striebel smoother for state estimation and sparse Bayesian learning for system identification. As such, we can alternately estimate hidden system states based on the learned LDSs and learn sparse system matrices  based on the estimated system states as shown in Figure 1 in the paper.
>
> > **The article "An Iterative Min-Min Optimization Method for Sparse Bayesian Learning" seems to be very related to the present article; specifically it does provide insights into why the priors on the weights promotes sparsity by comparing with regularizers employed. Given the earlier works, the contribution of the present article becomes more incremental.**
>
> * Note that the referenced paper focus on proposing a novel optimization method with theoretical guarantees to solve the problem formulated in Tipping's work. Hence, the models and problems addressed are still fundamentally different with ours.
> * In fact, our paper can also use the similar idea to empirically illustrate that why the priors imposed on system matrices can promote their sparsity. Without imposing sparsity-promoting priors on system matrices, the loss function derived for learning LDSs is as follows: $$Loss_1=H_1(A,B,R)+H_2(C,D,R),$$ where $H_1(A,B,R)$ and $H_2(C,D,R)$ are defined in the same way as in the paper. By introducing the sparsity-promoting priors, the loss function derived in the paper  includes an additional term $H_3(A,B,C,D,\Gamma_a,\Gamma_b,\Gamma_c,\Gamma_d)$: $$Loss_2=H_1(A,B,R)+H_2(C,D,R)+H_3(A,B,C,D,\Gamma_a,\Gamma_b,\Gamma_c,\Gamma_d).$$  By checking the explicit mathematical expression of $H_3(A,B,C,D,\Gamma_a,\Gamma_b,\Gamma_c,\Gamma_d)$ given in Appendix B1, we can find that it  includes the terms $\frac{A_{ij}^2}{\Gamma_{a,ij}},\frac{B^2_{ij}}{\Gamma_{b,ij}},\frac{C^2_{ij}}{\Gamma_{c,ij}},$ and $\frac{D^2_{ij}}{\Gamma_{d,ij}}$.  These terms are $\ell_2$ regularization terms that  penalize the squared values of the system matrices to promote their sparsity. Based on your comment, we will include this explanation to provide insights into why the priors on the weights promote sparsity in the revised version.

---

> ### Author Response · Authors · 2024-11-21
> **Rebuttal #2**
>
> **Response to Questions**
>
> > **Can the authors delineate the novelty with a comparison with the ideas presented in "Sparse Bayesian Learning and the Relevance Vector Machine" article by Tipping?**
>
> * See above.
>
> > **Can the authors state which aspects of this article differ in a substantive manner over analysis in "An Iterative Min-Min Optimization Method for Sparse Bayesian Learning"? Which new insights are found?**
>
> * As stated earlier, the models and problems addressed are fundamentally different. In their paper, they focus on using  sparse Bayesian learning to learn $y_t=\Phi(u_t)w+\varepsilon_t$ from input-output data. However, our paper considers the problem of learning
> $$x_t=Ax_{t-1}+Bu_t+\epsilon_t,\quad y_t=Cx_{t}+Du_t+\omega_t.$$ In particular, the hidden system state $x_t$ makes it difficult to learn system matrices. To address this issue, we explore the expectation-maximization framework to seamlessly integrate the strengths of the Rauch–Tung–Striebel smoother for state estimation and sparse Bayesian learning for system identification. In the expectation step, we use the Rauch–Tung–Striebel smoother to estimate the distribution of $x_t$ given $A,B,C$ and $D$. In the maximization step, we use the sparse Bayesian learning technique to learn sparse $A,B,C$ and $D$  given the distribution of $x_t$.
> *  Based on your previous comment, we provide insights into why the priors on system matrices can promote  their sparsity from a regularization standpoint. In addition, we also offer insights into why sparsity-promoting priors can preserve the inherent topological
> structure among the variables compared to classical algorithms as discussed in Section 4.1.
>
> > **Can the authors indicate why the priors promote structure of the matrices in contrast to sparsity alone?**
>
> * By applying the similarity transformation with any nonsingular matrix $P$, we can derive an equivalent realization of the original LDS with the system matrices being $PAP^{-1},PB,CP^{-1},$ and $D$, respectively. Hence, the classical algorithms only learn LDSs up to a similarity transformation, and the topological structure of the learned system matrices may differ entirely from that of the original. By employing sparsity-promoting priors, the proposed algorithm learns system matrices by balancing model complexity and modeling error. As such, the sparsity-promoting priors typically restrict the nonsingular matrix $P$ to be a generalized permutation matrix to avoid increasing model complexity, as this matrix only scales the values of the system matrices or permutes their rows or columns.  Hence, the proposed algorithm can maximally preserve the inherent topological structure among the variables as discussed in Section 4.1.

---

> ### Comment · Reviewer_Kt5E · 2024-11-21
> **Final responese**
>
> This reviewer remains unconvinced about the innovation, novelty and impact from prior state-of-the art. The specific problem is different; however, the main gist of the ideas remain incremental. I will stick with my earlier rating.

---

> ### Author Response · Authors · 2024-11-23
> **Rebuttal #3**
>
> We appreciate the reviewer for the prompt feedback. We would like to further clarify the contributions and innovations presented in our study.
>
>   1. Basically, the main difference between our paper and the referenced papers is whether the considered systems involve hidden system states.  Indeed, learning systems with the form $x'=f(x,u)+\varepsilon$ **[** or equivalently $y=f(u)+\varepsilon$ **]** from input-output data $(u_t,x_t)^T_{t=1}$  **[** or $(u_t,y_t)^T_{t=1}$ **]** using sparse Bayesian learning is a  well-studied problem, as the system state $x$ is  observable.   However, it should be highly noted that the presence of hidden system states makes it particularly difficult to learn systems from input-output data.  For systems involving hidden states of the form: $$ x'=f(x,u)+\varepsilon,$$ $$y=g(x,u)+\omega_t,$$
> **_Nature_**  published a paper to learn the hidden system sate $x$ and $f(x,u)$  from input-output data $(u_t,y_t)^T_{t=1}$ recently (Course & Nair, 2023), where the sparsity-inducing prior is also imposed on $f(x,u)$ to promote its sparsity.  However, they assume **$g(x,u)$ is known**, which is generally unrealistic.  Here, we address this issue under the linear case: $$ x_{t}=Ax_{t-1}+Bu_t+\varepsilon_t,$$ $$y_t=Cx_t+Du_t+\omega_t.$$
> Given input-output data $(u_t,y_t)^T_{t=1}$, our paper explores expectation-maximization framework to alternately leverage the Rauch–Tung–Striebel smoother to estimate hidden system state $x_t$ and sparse Bayesian learning to learn sparse system matrices $A,B,C,$ and $D$.  Although our study focuses on linear systems, we believe that the proposed algorithm offers valuable insights and methodologies with the potential to be extended to tackle nonlinear cases.
>
>    2. In this paper, our primary goal is not to propose a new state estimation method or a new sparse Bayesian learning algorithm. Instead, we propose an algorithm that unifies state estimation and sparse Bayesian learning by exploring the Expectation-Maximization (EM) framework. As such, we can learn LDSs with sparse system matrices when hidden system states are involved. We believe this integration is highly innovative and fundamentally distinct from traditional sparse Bayesian learning approaches.
>
>    **Reference**
>
> Kevin Course and Prasanth B Nair. State estimation of a physical system with unknown governing equations. Nature, 622(7982):261–267, 2023.

---

### Official Review · Reviewer_4Hwa · 2024-11-03

**Soundness:** 3
**Presentation:** 3
**Contribution:** 4
**Rating:** 6
**Confidence:** 4

**Summary:**

The paper discusses learning linear dynamical systems (LDSs) with sparse system matrices. The authors propose an algorithm to learn LDSs with sparse system matrices from noisy observation. The proposed algorithm applies the EM algorithm to give the MAP estimates of hidden states and system matrices. The derivative rule of structured matrices ensures the symmetry of the noise covariance matrix during the optimization process. The proposed algorithm outperforms the classical learning methods through the experimental results and simulation on the real-world dataset.

**Strengths:**

The proposed algorithm is more useful for exploring the interacting laws of the LDSs than the classical ones.

**Weaknesses:**

The proposed algorithm cannot determine the order n of the system from data directly and the similarity transformation may shrink many parameters to very small values, leading to numerical errors.

**Questions:**

1. The examples are simple, and the sparse matrices are diagonal. How are the sparse matrices selected in the problem?
2. Can the authors provide examples of using sparse matrices but not diagonal?

---

> ### Author Response · Authors · 2024-11-21
> **Rebuttal #1**
>
> We thank the reviewer for carefully reviewing our manuscript and for recognizing that the proposed algorithm is more useful for exploring the interacting laws of the LDSs than the classical ones. We hope to resolve your remaining concerns in our response.
>
> **Response to Weaknesses**
> > **The proposed algorithm cannot determine the order $n$ of the system from data directly and the similarity transformation may shrink many parameters to very small values, leading to numerical errors.**
>
> Based on your comment, we would like to further clarify the issues.
>
>   * While the proposed method cannot directly determine the order $n$ of the system from data, we can compare the performance of the learned systems across different orders and select the best one, as discussed in Section 6. In this context, we provide an empirical method to address this challenge.
>   * It is a well-known property of linear dynamical systems (LDSs) that the similarity transformation preserves the system's dynamics while altering the representation of the system matrices. Therefore, numerical errors arising from the similarity transformation are a common issue encountered by all learning algorithms for LDSs, regardless of the specific approach used.

---

> ### Author Response · Authors · 2024-11-21
> **Rebuttal #2**
>
> **Response to Questions**
> > **The examples are simple, and the sparse matrices are diagonal. How are the sparse matrices selected in the problem?**
>
> * While the simulated system in Section 5.1 is relatively simple,  it serves as an effective platform to both illustrate the comparative performance of our proposed algorithm and highlight its ability to capture the system's intrinsic structure. Specifically:
>   1. By considering a simple system, we are able to visually compare the system matrices learned by our algorithm with those obtained through classical algorithms, as shown in Table 1.
>   2. In addition, we aim to demonstrate that the proposed algorithm can preserve the inherent topological structure among variables when the considered LDSs adhere to the _Occam's razor_ principle. By considering a simple two-dimensional LDS with sparse and diagonal system matrices, we can easily construct a system following the _Occam's razor_ principle,  as the rank of the system matrices equals the number of nonzero components, as discussed in Section 4.1.
>
> * In Section 5.1, we select the system matrices to be sparse and diagonal in order to construct a system that follows the _Occam's razor_ principle. As such, we can verify that the proposed algorithm can preserve the inherent topological structure among variables by balancing model complexity and modeling error.
>
>
> > **Can the authors provide examples of using sparse matrices but not diagonal?**
>
> * Based on your valuable comment, we further test all the algorithms on sparse and non-diagonal system matrices. For the visual comparison of the learned system matrices, we still consider a two-dimensional LDS. Specifically, the experimental setup remains the same as described in Section 5.1, with the exception that we replace the system matrices with the following:
> $$
> A=\begin{bmatrix}
> 0 & 0.9 \\\\
> 0.9 & 0
> \end{bmatrix},
> B=\begin{bmatrix}
> 2 & 2 \\\\
> 2 & 0
> \end{bmatrix},
> C=\begin{bmatrix}
> 2 & 2\\\\
> 2 & 0
> \end{bmatrix},
> D=\begin{bmatrix}
> 1.5 & 1.5\\\\
> 1.5 & 0
> \end{bmatrix}.
> $$ The  table below  shows the learned system matrices of all the algorithms and the corresponding mean relative error (MRE) between the predicted outputs and real ones.
> | Method | Ours | PEM |4SID |MLE |
> |:--------|:--------|:--------|:--------|:--------|
> | $A$ | $$\begin{bmatrix} 0 &1.803\\\\ 0.447 & 0 \end{bmatrix} $$ |$$\begin{bmatrix} -0.548 &0.143\\\\ 3.561& 0.546 \end{bmatrix} $$|$$\begin{bmatrix} 0.895 &-0.027\\\\  -0.223& -0.894 \end{bmatrix} $$|$$\begin{bmatrix} -0.044 &1.881\\\\ 0.428& 0.045 \end{bmatrix} $$ |
> | $B$ | $$\begin{bmatrix} 8.179& 0\\\\ 4.121 &4.021 \end{bmatrix} $$ | $$\begin{bmatrix} 0.752& 7.824\\\\ 6.450& -15.217 \end{bmatrix} $$   | $$\begin{bmatrix} 0.005 &0.004\\\\ -0.012& 0 \end{bmatrix} $$  |$$\begin{bmatrix} 8.197& 2.749\\\\ 4.123& 2.566 \end{bmatrix} $$|
> | $C$|$$\begin{bmatrix} 0.494 &0.999\\\\ 0 &1.000 \end{bmatrix} $$  |$$\begin{bmatrix} 2.195& 0.891\\\\ 0.909& 0.464 \end{bmatrix} $$  | $$\begin{bmatrix} 1238.788 &-18.599\\\\ 604.052 &-140.493 \end{bmatrix} $$ |$$\begin{bmatrix} 0.475& 1.041\\\\ -0.587& 2.165\end{bmatrix} $$ |
> | $D$ |$$\begin{bmatrix} 1.379& 1.594\\\\ 1.422& 0\end{bmatrix} $$ | $$\begin{bmatrix} 9.570 &5.620\\\\ 5.550& 4.048\end{bmatrix} $$ | $$\begin{bmatrix} 0& 0\\\\ 0& 0\end{bmatrix} $$ | $$\begin{bmatrix} 1.378 &1.639\\\\ 1.433 &0.104 \end{bmatrix} $$ |
> | MRE | 4.38\% | 4.38\% |18.33\% |6.56\% |
>
>
>
>
>
>
>   In summary, the experimental results are consistent with those presented in the paper. The classical algorithms only learn system matrices up to a similar transformation. Hence, the learned systems are completely different with the original ones, which is less helpful for us to understand the interacting laws of  the system.  By balancing model complexity and modeling error, however, the proposed algorithm typically restricts the the nonsingular matrix $P$ of the similar transformation to be a generalized permutation matrix. As such, the learned LDS of the proposed algorithm generally preserves the topological structure of the system, differing only in terms of numerical values. In addition, this discrepancy is solely due to the scaled definition of the hidden system states, rather than a failure to capture the underlying system dynamics.  By comparing the learned $B$ and $C$ of the proposed algorithm with the real ones, we can derive $P\approx$[0 4; 2 0], which is consistent with theoretical analysis.  Consequently, the learned LDS of the proposed algorithm proves to be more useful for exploring the system's underlying interacting laws.

---

> > ### Author Response · Authors · 2024-12-02
> >
> > Dear Reviewer 4Hwa,
> >
> > Thank you for your thoughtful feedback.
> >
> > Please let us know whether our response addresses your concerns. If so, we kindly ask you to consider raising the rating of our work.
> >
> > We appreciate your time! We are looking forward to discussing any additional concerns you may have.
> >
> > Best wishes,
> >
> > Authors

---

### Official Review · Reviewer_Z2KB · 2024-11-03

**Soundness:** 3
**Presentation:** 3
**Contribution:** 2
**Rating:** 6
**Confidence:** 3

**Summary:**

This work addresses the problem of identifying linear dynamical systems with sparse system matrices from noisy observations.
According to the authors, available algorithms lack the ability to incorporate sparsity constraints. In contrast, their proposed solution promotes sparsity by imposing a Student's t-distribution prior on the system matrices. Then, it uses an expectation-maximization approach to estimate the system matrices. This dichotomy is illustrated with synthetic and real data.
Finally, the authors have also identified and corrected an inaccurate derivative rule employed by other expectation-maximization approaches in this context.

**Strengths:**

*Originality*:
The problem formulation and subsequent elaborations and conclusions appear novel.

*Quality*:
The proposed solution is carefully devised to admit closed-form updates. With this goal in mind, all heuristics and approximations seem justified and well motivated.

*Clarity*:
The paper is well organized and the ideas the authors want to convey are clearly exposed.

*Significance*:
This work may shed a light on the problem of learning linear dynamical systems with sparse system matrices. It can also open new research lines in this regard.

**Weaknesses:**

**W1**. Section 2 could benefit from a literature review on sparse promoting methods. For example, what is the state-of-the-art of Lasso-like estimators? or, how do they compare to sparse Bayesian learning in this context?

**W2**. I think the heuristic method introduced in Section 3.2. is *block coordinate descent* (also known as alternating minimization) rather than *block coordinate gradient descent* (BCGD).
Please, see [1] for a definition of the BCGD method.

For example, using the paper notation, the update step in eq. (22) can be equivalently written as
$ \boldsymbol{A}^{k+1}_r \simeq \text{arg} \underset{\boldsymbol{A}_r}{\text{ max }} H(\boldsymbol{\Theta} | \boldsymbol{\Theta}^k) $.
That is, the approximated gradient in eq. (21) is not used to perform a gradient step; instead, it is set to zero to find a stationary point in closed form.

I believe this distinction is not only a technicality, but is also important to emphasize that one could have used BCGD without approximating the gradient to find the stationary point numerically.

*Minor comments*:

**C1**. The main text does not seem to refer to Figure 1 anywhere. I think the reader would appreciate some guidance about which ideas or explanations Figure 1 is supporting or complementing.

**C2**. The authors may consider change the notation of the normal probability distribution in eq. (7)  from **|** to **;** to be aligned with the standarized notation encouraged by ICLR. Similarly, the matrix $\boldsymbol{P}$ introduced in eq. (35) may be renamed to not overlap with the matrices $\boldsymbol{P}^k_t$ and $\boldsymbol{P}^k_{t,t-1}$ introduced in eq. (9) and (15), respectively.

**C3**. I think the manuscript could benefit from an appendix where the derivative rule of structured matrices is explained, and why the derivatives of unstructured matrices cannot be used instead in general.

**C4**. I think Appendix C could benefit from explaining also how the mean and covariance of the noise $\bar{\epsilon}_t$ are obtained.

**References**:
[1] Beck, A., & Tetruashvili, L. (2013). On the convergence of block coordinate descent type methods. SIAM journal on Optimization, 23(4), 2037-2060.

**Questions:**

**Q1**. Section 3.2. is called ''Parameter and Hyperparameter Learning''.
Usually, the term hyperparameter is reserved for those parameters whose values control the learning process and cannot be infered during training (they can be learned by optimizing a higher level metric though).
Based on this definition, (I think) the only hyperparameters in this work are $a_0$ and $b_0$; however, they are not learned but initially set to a very small value.
So, by learning hyperparameters do you mean learning the parameters in the Gamma matrices? If so, do you call them hyperparameters because they are related to the hyperprior?

**Q2**. The last sentence of the Discussion section reads ''we hope to explore how to exactly learn LDSs with additional constraints'', and I am not sure what ''exactly'' or ''additional constraints'' refer to.
My best guess is that by exactly you mean maximizing eq. (6) in closed-form updates, and by additional constraints you mean additional constraints on top of sparsity?

---

> ### Author Response · Authors · 2024-11-21
> **Rebuttal #1**
>
> We thank the reviewer for carefully reviewing our manuscript and acknowledging our contributions. A point-by-point response to your comments follows below.
>
> **Response to Weaknesses**
> >**W1. Section 2 could benefit from a literature review on sparse promoting methods. For example, what is the state-of-the-art of Lasso-like estimators? or, how do they compare to sparse Bayesian learning in this context?**
>
> * Based on your constructive comment, we will give a literature review on sparsity-promoting methods in Section 2 in the revised version as follows:
>
>     **Sparsity-promoting methods.**  By adding a penalty term on model parameters, sparsity-promoting methods can balance model complexity and modeling error to learn systems from data. Leveraging the $\ell_1$ regularization term, Tibshirani (1996) proposes a method named Lasso to estimate parameters in linear models. Further, reweighted $\ell_1$ minimization methods are proposed to enhance sparsity (Wipf & Nagarajan, 2007; Candes et al., 2008). However, solving an $\ell_1$ minimization problem is challenging due to its non-differentiability at the origin, and these methods also require careful fine-tuning of hyperparameters. To address such issues, sparse Bayesian learning (SBL) imposes sparsity-promoting priors on model parameters to enforce sparsity. Subsequently, it efficiently maximizes the posterior distribution consisting of the likelihood function and prior to estimate model parameters and hyperparameters (Tipping, 2001; Wipf & Rao, 2004). Recently, SBL techniques have been applied to learn various systems from data (Pan et al., 2015; Yuan et al., 2019; Wang et al., 2024). However, leveraging SBL to learn LDSs with sparse system matrices remains a elusive and challenging problem due to hidden system states.
>
>     **References**
>
>     Robert Tibshirani. Regression shrinkage and selection via the Lasso. Journal of the Royal Statistica Society Series B: Statistical Methodology, 58(1):267–288, 1996.
>
>     David Wipf and Srikantan Nagarajan. A new view of automatic relevance determination. In Advances in Neural Information Processing Systems, volume 20, 2007.
>
>     Emmanuel J Candes, Michael B Wakin, and Stephen P Boyd. Enhancing sparsity by reweighted $\ell_1$ minimization. Journal of Fourier Analysis and Applications, 14:877–905, 2008.
>
>     Michael E Tipping. Sparse Bayesian learning and the relevance vector machine. Journal of Machine Learning Research, 1:211–244, 2001.
>
>     David P Wipf and Bhaskar D Rao. Sparse Bayesian learning for basis selection. IEEE Transactions on Signal Processing, 52(8):2153–2164, 2004.
>
>     Wei Pan, Ye Yuan, Jorge Gonc¸alves, and Guy-Bart Stan. A sparse Bayesian approach to the identification of nonlinear state-space systems. IEEE Transactions on Automatic Control, 61(1):182–187, 2015.
>
>     Ye Yuan, Xiuchuan Tang, Wei Zhou, Wei Pan, Xiuting Li, Hai-Tao Zhang, Han Ding, and Jorge Goncalves. Data driven discovery of cyber physical systems. Nature Communications, 10(1):4894, 2019.
>
>     Yasen Wang, Junlin Li, Zuogong Yue, and Ye Yuan. An iterative Min-Min optimization method for sparse Bayesian learning. In International Conference on Machine Learning, volume 235, 2024.
>
>
> > **W2. I think the heuristic method introduced in Section 3.2. is block coordinate descent (also known as alternating minimization) rather than block coordinate gradient descent (BCGD). Please, see [1] for a definition of the BCGD method. I believe this distinction is not only a technicality, but is also important to emphasize that one could have used BCGD without approximating the gradient to find the stationary point numerically.**
>
> * We really appreciate the reviewer for pointing out this mistake and feel sorry for it. Indeed, the method introduced in Section 3.2 is block coordinate descent (BCD) rather than block coordinate gradient descent (BCGD). We will revise the sentence accordingly in the updated version.
> * After carefully reviewing the BCGD method in [1], we found that applying it to the formulated problem in our paper faces two key issues:
>     1. The loss function in our paper is non-convex, and thus does not satisfy the basic assumption stated in [1], which requires the loss function to be a continuously differentiable convex function whose gradient is Lipschitz.
>      2. Since the loss function does not meet the assumption, we cannot determine the upper bounds on the Lipschitz constant, which is a necessary parameter for BCGD.
>
>     Consequently, we are unable to use BCGD to solve the formulated problem in our paper currently. Nonetheless, we recognize the potential value of this approach and plan to explore it further in future work.

---

> ### Author Response · Authors · 2024-11-21
> **Rebuttal #2**
>
> > **C1. The main text does not seem to refer to Figure 1 anywhere. I think the reader would appreciate some guidance about which ideas or explanations Figure 1 is supporting or complementing.**
>
> * We feel sorry for not referencing Figure 1 in the paper and will ensure it is properly cited in the revised version. In fact, Figure 1  gives the pipeline of the proposed algorithm as described in the third paragraph in Introduction. Specifically, this paper explores the expectation–maximization (EM) algorithm to give an alternate estimation of hidden system states and sparse system matrices. In the expectation step, we use the Rauch–Tung–Striebel (RTS) smoother to update the distribution of the hidden system states. In the maximization step, we leverage the BCD method to learn the sparse system matrices. By alternately performing the expectation and maximization steps until convergence, the proposed algorithm can determine the sparse system matrices of LDSs from noisy observations as shown in Figure 1.
>
> > **C2. The authors may consider change the notation of the normal probability distribution in eq. (7) from | to ; to be aligned with the standarized notation encouraged by ICLR. Similarly, the matrix  introduced in eq. (35) may be renamed to not overlap with the matrices
>  $P_t^k$ and $P_{t,t-1}^k$ introduced in eq. (9) and (15), respectively.**
>
> * Thank you for your valuable suggestions regarding the notation changes. Based on your feedback, we will revise the notations in the paper to enhance readability.
>
> >  **C3. I think the manuscript could benefit from an appendix where the derivative rule of structured matrices is explained, and why the derivatives of unstructured matrices cannot be used instead in general.**
>
> * We agree with the reviewer that the manuscript could benefit from such an appendix. Hence, we will include an appendix that highlights the differences between the derivative rules for structured and unstructured matrices as follows:
>      **Comparison of the derivative rules for structured and unstructured matrices**
>
>     Given the inherent symmetry of noise covariance matrices, we need to consider the derivative rule of structured matrices during the optimization process to guarantee their symmetry. However, many learning algorithms overlook this property and apply the derivative rule designed for unstructured matrices, which we argue is inappropriate when updating noise covariance matrices.
>
>     Consider a symmetric matrix $U\in \mathbb R^{n\times n}$ and a function $f(U): \mathbb R^{n\times n}\to \mathbb R$. If we mistakenly apply the derivative rule for unstructured matrices, the derivative of $f(U)$ with respect to $U$ is given as  $\frac{df(U)}{dU}$. However, the correct derivative expression for the symmetric matrix $U$ is $\frac{df(U)}{dU}+\left(\frac{df(U)}{dU}\right)'-\textup{diag}\left[\frac{df(U)}{dU}\right]$ (Petersen et al., 2008). Hence, we cannot use the derivative of unstructured matrices to update  noise covariance matrices during the optimization process due to their inherent symmetry.
>
>     **References**
>
>     Kaare Brandt Petersen, Michael Syskind Pedersen, et al. The matrix cookbook. Technical University of Denmark, 7(15):510, 2008.
>
>
> > **C4. I think Appendix C could benefit from explaining also how the mean and covariance of the noise $\overline \epsilon_t$ are obtained.**
>
> * Based on your valuable comment, we will explain how to obtain the mean and covariance of  $\overline \varepsilon_t$ in Appendix C in the revised version as follows:
>
>     Because $\varepsilon_t\sim N(0,R)$ and $\overline \varepsilon_t=P\varepsilon_t$, we can derive the mean of $\overline \varepsilon_t$ as follows:  $$\mathbb E[\overline \varepsilon_t]=\mathbb E[P\varepsilon_t]=P\mathbb E[\varepsilon_t]=0.$$ In addition, the covariance of $\overline \varepsilon_t$ can be derived as follows: $$ \mathbb E[\left(\overline \varepsilon_t-\mathbb E[\overline \varepsilon_t]\right)\left(\overline \varepsilon_t-\mathbb E[\overline \varepsilon_t]\right)']=\mathbb E[P\varepsilon_t\varepsilon_t'P'] =PE[\varepsilon_t\varepsilon_t']P'=PRP'.$$

---

> ### Author Response · Authors · 2024-11-21
> **Rebuttal #3**
>
> **Response to Questions**
> > **Q1. Section 3.2. is called ''Parameter and Hyperparameter Learning''. Usually, the term hyperparameter is reserved for those parameters whose values control the learning process and cannot be infered during training (they can be learned by optimizing a higher level metric though). Based on this definition, (I think) the only hyperparameters in this work are $a_0$
>  and $b_0$; however, they are not learned but initially set to a very small value. So, by learning hyperparameters do you mean learning the parameters in the Gamma matrices? If so, do you call them hyperparameters because they are related to the hyperprior?**
>
> * By learning hyperparameters, we are indeed referring to learning the parameters in the Gamma matrices. In sparse Bayesian learning, these quantities are additionally introduced to control the existing model parameters.  Therefore, they are termed hyperparameters regardless of whether a hyperprior is associated with them (Tipping, 2001; Wipf & Rao, 2004).
>
>     **References**
>
>     Michael E Tipping. Sparse Bayesian learning and the relevance vector machine. Journal of Machine Learning Research, 1:211–244, 2001.
>
>     David P Wipf and Bhaskar D Rao. Sparse Bayesian learning for basis selection. IEEE Transactions on Signal Processing, 52(8):2153–2164, 2004.
>
> > **Q2. The last sentence of the Discussion section reads ''we hope to explore how to exactly learn LDSs with additional constraints'', and I am not sure what ''exactly'' or ''additional constraints'' refer to. My best guess is that by exactly you mean maximizing eq. (6) in closed-form updates, and by additional constraints you mean additional constraints on top of sparsity?**
>
> * We apologize for the confusing phrasing. As discussed in Section 4, it is well-known that the classical learning algorithms for LDSs only learn system matrices up to a similar transformation. This transformation provides an equivalent realization of the original LDSs. However, it changes not only the numerical values but, more importantly, the topological structure of  system matrices, resulting in potential  misinterpretation of intrinsic working mechanisms. By considering the sparsity constraint on system matrices, the proposed algorithm learns LDSs by adopting  sparsity-promoting priors to balance model complexity and modeling error. Hence, the nonsingular matrix of the similar transformation is typically restricted to be a generalized permutation matrix to avoid increasing model complexity. Thus, the system matrices learned by the proposed algorithm can  preserve the topological structure of the true ones compared to the classical algorithms. However, they still differ in terms of numerical values due to the scaled definitions of system states. Hence, our future work hope to explore how to ensure that the learned system matrices are exactly the same as the true ones. To this end, we may need to consider additional constraints on system matrices beyond the sparsity constraint.
>
>     Based on your comment, we will revise this sentence in the updated version as follows:
>
>      _In our future work, we hope to  explore how to ensure that the learned system matrices are exactly the same as the true ones by imposing additional constraints on system matrices beyond the sparsity constraint._

---

> > ### Comment · Reviewer_Z2KB · 2024-11-24
> >
> > Thank you for iterating over my concerns and resolving my questions.
> > I feel confirmed in my decision and I keep my rating.

---

> > > ### Author Response · Authors · 2024-11-25
> > >
> > > Thank you for your positive feedback and for confirming your decision. We appreciate your time and thoughtful comments throughout the review process.

---

### Official Review · Reviewer_ukAS · 2024-11-04

**Soundness:** 2
**Presentation:** 3
**Contribution:** 2
**Rating:** 5
**Confidence:** 3

**Summary:**

Proposing a Bayesian algorithm for model parameter estimation in dynamical systems with sparse components.

**Strengths:**

1. Good numerical performance (Table 2).

2. Enough details on the proposed algorithm.

**Weaknesses:**

1. Novelty is rather limited which questions the overall contribution of the manuscript.

2. Many missing papers in the literature review such as:

- Faradonbeh, M. K. S., Tewari, A., & Michailidis, G. (2018). Finite time identification in unstable linear systems. Automatica, 96, 342-353.

- Chakraborty, N., Khare, K., & Michailidis, G. (2023). A Bayesian framework for sparse estimation in high-dimensional mixed frequency vector autoregressive models. Statistica Sinica, 33, 1629-1652.

- Faradonbeh, M. K. S., Tewari, A., & Michailidis, G. (2020). Optimism-based adaptive regulation of linear-quadratic systems. IEEE Transactions on Automatic Control, 66(4), 1802-1808.

-Samanta, S., Khare, K., & Michailidis, G. (2022). A generalized likelihood-based Bayesian approach for scalable joint regression and covariance selection in high dimensions. Statistics and computing, 32(3), 47.

- Modi, A., Faradonbeh, M. K. S., Tewari, A., & Michailidis, G. (2024). Joint learning of linear time-invariant dynamical systems. Automatica, 164, 111635.


3. The real data section can be improved by further discussions; so far, only improved prediction errors are reported.

**Questions:**

1. How sensitive Algorithm 1 is with respect to initial values (initial guesses of model parameters)?

2. Numerical examples are 2-3 dimensional. How does the proposed algorithm work under high-dimensional settings? Sparsity assumption typically makes sense in such high-dimensional cases.

3. What can be said about inference for model parameters?

---

> ### Author Response · Authors · 2024-11-21
> **Rebuttal #1**
>
> We thank the reviewer for taking the time to review our paper. In addition, we hope to address your concerns and questions here:
>
> **Response to Weaknesses**
>
> > **Novelty is rather limited which questions the overall contribution of the manuscript.**
>
> * We understand your concerns regrading the novelty of our work, and would like to take this opportunity to further clarify the contributions and innovations presented in our study.
>     1.  Many linear dynamical systems (LDSs) have sparse system matrices because interactions among variables are limited or only a few significant relationships exist. While learning LDSs is a well-studied problem, existing methods only learn system matrices of LDSs up to a similarity transformation. This transformation provides an equivalent realization of the original LDSs but often alters not only the numerical values of the system matrices but also their topological structure. Hence, available learning algorithms lack the ability to learn system matrices with the sparsity constraint. To address this issue, this paper presents a novel method to learn LDSs by imposing sparsity-promoting priors on system matrices.  The method effectively balances model complexity and modeling error during the optimization process. Therefore, the nonsingular matrix in the similarity transformation is typically constrained to be a generalized permutation matrix to avoid increasing model complexity, as this matrix only scales the values of the system matrices or permutes their rows or columns. As a result, the learned LDSs generally exhibit the same topological structure  as the real ones,  which is a significant improvement over traditional methods. This property makes our method  particularly valuable for revealing the system's internal structure and interaction rules.
>     2.   Because system states are unknown, learning LDSs with sparse system matrices is particularly challenging. Specifically, the derived loss function is non-convex and even does not have an explicit form due to hidden system states. To address this issue, we propose an optimization algorithm by exploring expectation-maximization framework. This algorithm alternately leverages the Rauch–Tung–Striebel smoother to estimate hidden system states and  sparse Bayesian learning to identify sparse system matrices.  Experimental results on both simulated and real-world datasets demonstrate that the proposed algorithm outperforms the classical ones on learning LDSs with sparse system matrices.
>
> > **Many missing papers in the literature review.**
>
> * We appreciate the reviewer for highlighting these papers that are closely related to our work. We assure the reviewer that these papers will be appropriately cited in the revised version.
>
> > **The real data section can be improved by further discussions; so far, only improved prediction errors are reported.**
>
> * We agree that further discussions on the real data section would enhance the depth of our findings. Based on your comment, we will provide a more detailed analysis in the revised version as follows:
>
>     _Table 2 displays the mean relative error (MRE) between the predicted outputs of all the learned LDSs and real ones. Due to the lack of the ground truth, the learned system matrices of all the algorithms are not depicted for comparison. Because the proposed algorithm obtains minimum MRE on both datasets, experimental results demonstrate its superiority over classical algorithms. In addition, note that the classical algorithms only learn LDSs up to a similarity transformation, which may lead to misinterpretations of intrinsic working mechanisms.  However, the proposed algorithm learns system matrices by balancing model complexity and modeling error. As a result, the learned system matrices typically preserve the inherent topological structure among the variables, which is more valuable for exploring the interaction laws of systems._

---

> ### Author Response · Authors · 2024-11-21
> **Rebuttal #2**
>
> **Response to Questions**
>
> > **How sensitive Algorithm 1 is with respect to initial values (initial guesses of model parameters)?**
>
> * In the experiments, the initial values of $A$, $B$, $C$, $D$, $R$, and $Q$ are all set as identity matrices. To evaluate the sensitivity of Algorithm 1, we further test the proposed algorithm on the system described in Section 5.1 by initializing the model parameters as $aI_2$, where $a$ varies from 0.5 to 1.5 in  increments of 0.1. The following table records whether the learned system matrices of the proposed algorithm preserve the topological structure of the true ones and the mean relative error (MRE) defined in the paper:
>
>     | $a$ | 0.5 | 0.6 |0.7 | 0.8|0.9 | 1 |1.1 | 1.2 |1.3 | 1.4 |1.5|
>     |:--------|:--------:|:--------:|:--------:|:--------:|:--------:|:--------:|:--------:|:--------:|:--------:|:--------:|:--------:|
>     |  Topology Preservation   |   $\times$     |   $\checkmark$         |$\checkmark$     | $\checkmark$     | $\checkmark$     | $\checkmark$     | $\checkmark$     | $\checkmark$     | $\checkmark$     | $\times$     | $\times$     |
>     |  MRE   |   0.0787    |    0.0737    | 0.0737    | 0.0737    | 0.0737    | 0.0737    | 0.0737    | 0.0737    | 0.0737    |0.0737    |0.0737  |
>
>   As observed from the table, the experimental results are consistent with those in the paper when $a$ is in the range [0.6,1.3], indicating that the proposed algorithm is relatively robust to initial values. Remarkablely, even when the initial state transition matrix $A$ is unstable (i.e., $a>1$), the proposed algorithm is still able to accurately learn the topological structure of the true systems.
>
> > **Numerical examples are 2-3 dimensional. How does the proposed algorithm work under high-dimensional settings? Sparsity assumption typically makes sense in such high-dimensional cases.**
>
> * We agree with the reviewer that the sparsity assumption typically makes sense in high-dimensional cases. Based on your constructive comment, we further test all the algorithms on a 15-dimensional system. As a result, each system matrix has 225 unknown parameters. Specifically, we consider $A$ to be an anti-diagonal matrix with nonzero components equal to 0.9 and set $B=C=D=1.5I_{15}$ and $R=Q\sim \mathcal N(0,0.81I_{15})$. Because these system matrices are extremely sparse,  accurately learning their topological structures is quite challenging. As for algorithm implementation, we collect 2400 data points and initialize $A, B, C, D, R,$ and $Q$ as identity matrices.  The learned parameters less than the
> threshold 0.002 are removed from the result.  The following table records whether the learned system matrices  preserve the topological structure of the true ones and the mean relative error (MRE) defined in the paper:
> | Method | Ours |  PEM | 4SID | MLE |
> |----------|----------|----------|----------|----------|
> | Topology Preservation |  $\checkmark$ | $\times$ | $\times$ | $\times$ |
> | MRE |  **13.66\%**|  19.96\% | 25.99\% | 13.91\%|
>
>     Obviously, the experimental results on high-dimensional systems are consistent with those observed in low-dimensional systems. The proposed algorithm preserves the inherent topological structure  among the variables and achieve the lowest MRE.

---

> > ### Author Response · Authors · 2024-12-02
> >
> > Dear Reviewer ukAS,
> >
> > Thank you for your thoughtful feedback.
> >
> > Please let us know whether our response addresses your concerns. If so, we kindly ask you to consider raising the rating of our work.
> >
> > We appreciate your time! We are looking forward to discussing any additional concerns you may have.
> >
> > Best wishes,
> >
> > Authors

---

> > > ### Comment · Reviewer_ukAS · 2024-12-02
> > >
> > > Thank you for providing additional details. I still have some doubts on the overall novelty/contribution of the work while responses to Q2 and 3 are incomplete: for Q2, I expect a high-dimensional case, but authors just increased the dimension a bit while the sample size is selected to be very large. This is not a high-dim case; for Q3, there seems to be no response related to providing inferential framework. Thus, I will keep my current rating for this work.

---

> > > > ### Author Response · Authors · 2024-12-03
> > > > **Rebuttal #4**
> > > >
> > > > We appreciate the reviewer for the prompt feedback.
> > > >
> > > > > **Novelty/Contribution**
> > > >
> > > > *  It should be highly noted that learning systems with sparsity constraints and hidden system states is particularly challenging. For the system involving hidden system states of the form: $$x'=f(x)+\varepsilon,$$ $$y=g(x)+\omega.$$ **_Nature_** recently published a paper to estimate $x$ and learn $f(x)$  from collected data (Course & Nair, 2023). However, they still need to assume that $g(x)$  is known, which is generally unrealistic. Here, we address this issue under the linear case: $$x_t=Ax_{t-1}+Bu_t+\varepsilon_t,$$$$y_t=Cx_t+Du_t+\omega_t$$
> > > > Our paper explores expectation-maximization framework to alternately leverage the Rauch–Tung–Striebel smoother to estimate hidden system state $x_t$ and sparse Bayesian learning to learn sparse system matrices $A,B,C,$ and $D$. Although our study focuses on linear systems, we believe that the proposed algorithm offers valuable insights and methodologies with the potential to be extended to tackle nonlinear cases.
> > > >
> > > >     **Reference**
> > > >
> > > >     Kevin Course and Prasanth B Nair. State estimation of a physical system with unknown governing equations. Nature, 622(7982):261–267, 2023.
> > > >
> > > > > **Q2**
> > > >
> > > > * The sample size used for the higher-dimensional system (i.e., 2400) is approximately the same as that for the low-dimensional case (i.e., 2000) in the paper. We did not deliberately choose a very large sample size. In addition, because we consider a 15-dimensional system, each system matrix has $225=15\times 15$ unknown parameters. Consequently, the system matrices contain a total of 900 parameters that need to be estimated. Thus, the selected sample size is reasonable.
> > > >
> > > >
> > > > * Because the proposed algorithm involves the inversion of multiple $n\times n$ matrices, where $n$ is the system dimension,  the computational cost scales with $n^3$. Hence, we need to admit that a limitation of the proposed algorithm  is its high computational complexity. However, the experimental results on  the 15-dimensional system have demonstrated its potential to handle high-dimensional cases when sufficient computational resources are available.
> > > >
> > > > * We hope the reviewer could understand  the challenge involved in applying the proposed algorithm to very high-dimensional systems. For example, in the case of a 100-dimensional system, there are 40000 unknown parameters that need to estimated. To  ensure the performance of the proposed algorithm, we need to increase the sample size accordingly, making the experiment extremely time-consuming.
> > > >
> > > > > **Q3**
> > > >
> > > > * For the state-space representation of LDSs,  the proposed  inferential framework for the first time provides a unified approach for state estimation and sparse system identification. Specifically, this paper explores the expectation–maximization (EM) algorithm to give an alternate estimation of hidden system states and sparse system matrices. In the expectation step, we use the Rauch–Tung–Striebel (RTS) smoother to update the distribution of hidden system states given the learned system matrices. In the maximization step, we leverage the sparse Bayesian learning to learn the sparse system matrices given the updated distribution of system states.  By alternately performing the expectation and maximization steps until convergence, the proposed algorithm can determine the sparse system matrices and system states of LDSs.
> > > > * In addition, we believe that the proposed inferential framework offers valuable insights and methodologies with the potential to be extended to tackle nonlinear cases as discussed earlier, which would inspire many follow-up works.

---

> ### Author Response · Authors · 2024-11-21
> **Rebuttal #3**
>
> > **What can be said about inference for model parameters?**
>
> * In this paper, we consider the state-space representation of LDSs, which involves two fundamental equations: the state transition equation $x_t=Ax_{t-1}+Bu_t+\varepsilon_t$ and the observation equation $y_t=Cx_t+Du_t+\omega_t$. Because the system state $x_t$ is unknown, it is  difficult to learn system matrices $A, B, C,$ and $D$ with the sparsity constraint from input-output data $(u_t,y_t)_{t=1}^T$. To address this issue, this paper explores the expectation-maximization framework to alternately leverage the  Rauch–Tung–Striebel (RTS) smoother to estimate hidden system states and sparse Bayesian learning to identify spare system matrices. As such, we present a closed-form update rule to iteratively  learn $A,B,C,$ and $D$ until  convergence. Experimental results on both simulated and real-world datasets demonstrate that  the proposed algorithm outperforms the classical ones on learning LDSs with sparse system matrices.
>
> * As stated earlier, existing methods only learn system matrices of LDSs up to a similarity transformation. However, the proposed algorithm introduces sparsity-promoting priors to balance model complexity and modeling error to learn system matrices. Hence, The learned system matrices of the proposed algorithm typically preserve the topological structure of the true ones compared to the classical algorithms. While they still differ in terms of numerical values, this discrepancy is solely due to the scaled definition of the hidden system states, rather than a failure to capture the underlying system dynamics. This property makes the inferred  model parameters particularly valuable for providing interpretable insights into the interaction laws between variables.

---

### Official Review · Reviewer_Ayif · 2024-11-04

**Soundness:** 3
**Presentation:** 3
**Contribution:** 2
**Rating:** 3
**Confidence:** 4

**Summary:**

The authors consider learning of linear dynamical systems with a sparsity-promoting regularization term. When one considers the learning of linear dynamical systems with side-constraints as a non-convex optimization problem, one can quite naturally obtain solvers the converge to global minimizers of the least-squares estimator, asymptotically. Even the 2020 paper appearing in SIAM Review (https://proceedings.mlr.press/v120/ahmadi20a) considers a variety of types of side information, rather than just sparsity, and the more recent papers provide detailed analyses (e.g. non-asymptotic error bounds in the Frobenius norm in https://hal.science/hal-04394297/document), or consider operator-valued formulations of the problem (e.g., https://doi.org/10.1109/TAC.2023.3313351). The present paper does not do any of this, focussing on an EM heuristic.

**Strengths:**

The results are reasonably well written up.

**Weaknesses:**

The authors obtain an EM heuristic for the problem, but do not prove whether the EM heuristic provides good solutions or not.

The authors test their approach on two instances from "Database for the Identification of Systems" (http://homes.esat.kuleuven.be/~smc/daisy/), a well-known collection of 20+ instances, without explaining why they choose the two instances in particular. This smacks of cherry-picking. The comparison against PEM and n4sid, which date back to 1970s and 1994, respectively, also does not seem to capture the current state of the art fully.

The authors do not consider any of the recent literature on learning of linear dynamical systems with side-constraints as a non-convex optimization problem (e.g. https://proceedings.mlr.press/v120/ahmadi20a, https://doi.org/10.1109/TAC.2023.3313351, https://hal.science/hal-04394297/document).

**Questions:**

Would you have results on further instances from "Database for the Identification of Systems"?

Would you have a comparison against any recently proposed system ID methods? There is the conference series L4DC (e.g. https://l4dc.web.ox.ac.uk/home), where a dozen a presented each year, for instance.

---

> ### Author Response · Authors · 2024-11-21
> **Rebuttal #1**
>
> We would like to express our gratitude to the reviewer for your valuable time and efforts in reviewing our work.
>
> **Response to Summary**
>
> * We appreciate the reviewer pointing out the papers on learning linear dynamical systems (LDSs), and we have carefully reviewed them in light of the comment, which enable us to further clarify our work. Basically, the referenced papers only consider **special forms of LDSs** with tractable side-constraints and thus can convert the LDS learning problem into a **convex optimization problem**. As such, they can use off-the-shelf solvers to address the problem efficiently and provide theoretical analysis under certain conditions. More specifically:
>
>     1. For the paper in https://proceedings.mlr.press/v120/ahmadi20a, their interest is to learn a dynamical system $x’=f(x)$ subject to side information. Using polynomial functions to approximate $f(x)$, they can leverage ordinary least squares method to estimate the corresponding coefficients from data, leading to a convex quadratic function. In addition, they can impose six types of side information (do not include sparsity) on polynomial vector fields in a numerically tractable fashion. As such, they are able to develop semidefinite programming based approach to solve the formulated convex optimization problem.
>
>     2. For the paper in https://hal.science/hal-04394297/document, they consider the problem of learning LDSs of the form $x_{t+1}=Ax_t +\varepsilon_t$ with the assumption that $A$ belongs to a known convex set. As such, they can obtain $A$ by the constrained least squares estimator, which is also a convex program that can typically be solved efficiently in practice.
>
>     3. For the paper in https://doi.org/10.1109/TAC.2023.3313351, they focus on learning  LDSs with the form $$x_t=Ax_{t-1}+\varepsilon_t, y_t=Bx_t+\omega_t.$$ First, they cast the system learning problem as a noncommutative polynomial optimization problem (NCPOP). Under some conditions, they derive the semidefinite programming relaxation of the NCPOP, which is also a convex optimization problem.
>
>     However, our paper considers the most general form of LDSs as follows: $$x_t=Ax_{t-1}+Bu_t+\varepsilon_t, \quad y_t=Cx_t+Du_t+\omega_t.$$ In addition, we consider the sparsity constraint on system matrices.  As a result, the derived loss function is **non-convex**  and even has no analytical expression, as the likelihood function is hard to be explicitly computed.  Due to the complexity of the formulated problem, we are unable to develop an optimization algorithm with theoretical analysis to solve it  currently. As a result, we develop an EM-based heuristic method for estimating model parameters.  In addition, note that the experimental results on both simulated and real-world datasets demonstrate that the proposed algorithm outperforms the classical ones on learning LDSs with sparse system matrices.
>
> **Response to Weaknesses**
>
> > **The authors obtain an EM heuristic for the problem, but do not prove whether the EM heuristic provides good solutions or not.**
> * In this paper, we consider the problem of learning the most general form of LDSs with the sparsity constraint imposed on system matrices. Following the Bayes’ rule, we can combine the marginal likelihood function and sparsity-promoting prior to derive the posterior distribution of all the unknown parameters. According to the maximum a posteriori (MAP) principle, we can estimate the parameters by maximizing the posterior function. However, directly maximizing such a loss function is intractable because the marginal likelihood function is non-convex and is hard to be explicitly computed. To address this issue, we thus propose an EM heuristic for the problem.
>
> * We acknowledge the lack of theoretical results in the current work. This is primarily due to the complexity of the non-convex loss function and the challenges involved in developing a rigorous theoretical optimization framework at this stage. However, we believe that the empirical results provide strong evidence supporting the proposed method and can serve as a foundation for future theoretical investigations.

---

> > ### Comment · Reviewer_Ayif · 2024-12-01
> > **Response**
> >
> > Dear all,
> >
> > many thanks for the rebuttal.
> >
> > Notice that polynomial optimization problems are, in general, non-convex. The sole fact that you obtain a convex relaxation within solving the non-convex problem does not render the original problem any less non-convex. Even the multi-variate versions of learning the autonomous system are non-convex, and the uni-variate versions may exhibit invexity in some transfer-function (rather than state-space) formulation:
> > https://jmlr.org/papers/volume19/16-465/16-465.pdf
> >
> > I appreciate that the non-convexity does make the problem challenging -- but "sweeping it under the carpet" rather than tackling the challenge makes the problem less interesting.
> >
> > I would keep my score.

---

> > > ### Author Response · Authors · 2024-12-02
> > > **Rebuttal #4**
> > >
> > > We appreciate the reviewer for the prompt feedback.
> > >
> > > * By using polynomial functions to approximate $f(x)$,  the paper in https://proceedings.mlr.press/v120/ahmadi20a learns the coefficients of polynomial functions via the least-squares loss, which is a well-known convex function.
> > > * For the multi-variate versions of learning the autonomous system, the paper in https://hal.science/hal-04394297/document also uses the least squares estimator to estimate the state transition matrix $A$. Hence, the formulated optimization problem is also convex, as stated in the paper (see equation 1.3).
> > > * In this paper, our primary focus is to propose an algorithm for learning the most general LDSs with sparse system matrices. While we are unable to provide a satisfactory theoretical analysis at this stage, we hope the reviewer can understand that it is not feasible to address all aspects in a single paper. In addition,  it is worth noting that many theoretical works are often built upon methods already proposed by others. Therefore, we  believe the proposed algorithm would inspire many follow-up works.

---

> ### Author Response · Authors · 2024-11-21
> **Rebuttal #2**
>
> > **The authors test their approach on two instances from "Database for the Identification of Systems" (http://homes.esat.kuleuven.be/~smc/daisy/), a well-known collection of 20+ instances, without explaining why they choose the two instances in particular. This smacks of cherry-picking. The comparison against PEM and n4sid, which date back to 1970s and 1994, respectively, also does not seem to capture the current state of the art fully.**
>
> * We totally understand your concern about potential cherry-picking, and we feel sorry for the lack of explanation regarding the selection criteria. Basically, the following criteria are involved in choosing the datasets:
>     1. Since Section 5.2 focuses on industrial process systems, the datasets used are specifically drawn from the "Process Industry Systems" subset of the Database for the Identification of Systems, rather than other categories.
>     2. To empirically ensure the performance of all the algorithms, we prefer to select datasets with at least 1,000 sample points.
>     3. To ensure system matrices have a sparse topology, the system typically needs to be multidimensional. Hence, datasets with one-dimensional inputs or outputs are excluded.
>
>     Therefore, although there are over 20 instances, we have only tested all the algorithms on a subset of the dataset. Based on your comment, we will give the selection criteria to explain why we choose the two instances in the revised version.
>
> * While PEM and n4sid methods trace back to the 1970s and 1994, respectively, they still remain leading techniques for learning LDSs because they consider the most general form of such systems. Hence, they are  regarded as standard approaches in system identification and are widely used as baselines in recent works (Hazan et al., 2017; Hazan et al., 2018; Zhou & Marecek,2023). Particularly, note that while there have been some recent articles on learning LDSs, many of these focus on special forms of LDSs rather than the general case (Mamakoukas et al., 2020; Jedra & Proutiere, 2020; Jongeneel et al.,
> 2022;Tyagi & Efimov, 2023). Consequently, we are unable to compare the proposed method with them in our experiments.  Thus, while PEM and n4sid are not the most recent methods, they still provide a solid benchmark for comparison in the context of learning the most general LDSs.
>
>     **References**
>
>     Elad Hazan, Singh Karan, and Zhang Cyril. Learning linear dynamical systems via spectral filtering. Advances in Neural Information Processing Systems, volume 30, 2017.
>
>     Elad Hazan, Holden Lee, Karan Singh, Cyril Zhang, and Yi Zhang. Spectral filtering for general linear dynamical systems. Advances in Neural Information Processing Systems, volume 31,  2018.
>
>     Quan Zhou and Jakub Marecek. Learning of linear dynamical systems as a non-commutative polynomial optimization problem. IEEE Transactions on Automatic Control, 2023.
>
>     Giorgos Mamakoukas, Orest Xherija, and Todd Murphey. Memory-efficient learning of stable linear dynamical systems for prediction and control. In Advances in Neural Information Processing Systems, pp. 13527–13538, 2020.
>
>     Yassir Jedra and Alexandre Proutiere. Finite-time identification of stable linear systems optimality of the least-squares estimator. In IEEE Conference on Decision and Control, pp. 996–1001, 2020.
>
>     Wouter Jongeneel, Tobias Sutter, and Daniel Kuhn. Efficient learning of a linear dynamical system with stability guarantees. IEEE Transactions on Automatic Control, 68(5):2790–2804, 2022.
>
>     Hemant Tyagi and Denis Efimov. Learning linear dynamical systems under convex constraints.  arXiv preprint arXiv:2303.15121, 2023.
>
> > **The authors do not consider any of the recent literature on learning of linear dynamical systems with side-constraints as a non-convex optimization problem (e.g. https://proceedings.mlr.press/v120/ahmadi20a, https://doi.org/10.1109/TAC.2023.3313351, https://hal.science/hal-04394297/document).**
>
> * As discussed in the response to Summary, while the referenced papers can learn LDSs, they only consider special forms of LDSs. Hence, we cannot apply them to learn the LDSs with the most general form in our paper and thus cannot compare the proposed algorithm with them in experiments.
> * Given simplified forms, note that the referenced papers actually formulate a convex optimization problem to learn LDSs with tractable side-constraints. As such, they are able to use off-the-shelf solvers to solve the problem efficiently. However, how to apply these techniques to learn the most general form of LDSs remains elusive and challenging, even without considering the sparsity constraint on system matrices.

---

> > ### Comment · Reviewer_Ayif · 2024-12-01
> > **Response**
> >
> > While I agree that some of the references consider autonomous LDS or observable LDS or similar, which may be "special forms of LDSs", the methods proposed would be easy to extend to the controlled, partially observable case the authors wish to study?

---

> > > ### Author Response · Authors · 2024-12-02
> > > **Rebuttal #5**
> > >
> > > Indeed, the references provide very nice results for learning special forms of LDSs. However, we respectfully disagree with the view that they can be extended to the formulated problem in our paper. It is widely acknowledged that learning systems with sparsity constraints and hidden states is particularly challenging.
> > >
> > > For example,  while learning $x'=f(x)+\varepsilon$  from collected data is a well-studied problem, learning $f(x)$ with the sparsity constraint when the system involves hidden states remains elusive. For the system involving hidden system states of the form:
> > >      $$x'=f(x)+\varepsilon, $$ $$y=g(x)+\omega.$$
> > >  **_Nature_** recently published a paper to estimate  $x$ and learn $f(x)$ from collected data (Course & Nair, 2023). However, they still need to assume that $g(x)$ is known, which is generally unrealistic. Here, we address this issue under the linear case:
> > > $$x_{t}=Ax_t+Bu_t+\varepsilon,$$ $$y_t=Cx_t+Du_t+\omega.$$
> > > Our paper explores expectation-maximization framework to alternately leverage the Rauch–Tung–Striebel smoother to estimate hidden system state $x_t$ and sparse Bayesian learning to learn sparse system matrices $A,B,C,$ and $D$. Although our study focuses on linear systems, we believe that the proposed algorithm offers valuable insights and methodologies with the potential to be extended to tackle nonlinear cases.
> > >
> > >
> > > **Reference**
> > >
> > > Kevin Course and Prasanth B Nair. State estimation of a physical system with unknown governing equations. Nature, 622(7982):261–267, 2023.

---

> ### Author Response · Authors · 2024-11-21
> **Rebuttal #3**
>
> > **Would you have results on further instances from "Database for the Identification of Systems"?**
> * To address your concern about potential cherry-picking, we further test all the algorithms on a dataset that adheres to the above selection criteria. The data comes from a model of a Steam Generator at Abbott Power Plant in Champaign IL, including four inputs and  four outputs with a length of 9600. The  table below records  the mean relative error  (MRE) between the predicted outputs of all the learned LDSs and real ones:
>
>     | Method | Ours |  PEM | 4SID| MLE |
>     |----------|----------|----------|----------|----------|
>     | MRE | **21.45\%** | 22.70\% | 29.26\% | 22.23\% |
>
>     Obviously, the experimental results are consistent with those reported in the paper.
>
> > **Would you have a comparison against any recently proposed system ID methods? There is the conference series L4DC (e.g. https://l4dc.web.ox.ac.uk/home), where a dozen a presented each year, for instance.**
>
> * Based on your comment, we further compare the proposed algorithm with the method presented in Oymak & Ozay (2019). We are deeply grateful to the authors for sharing their code with us. Note that while addressing the learning problem for the most general LDSs, they did not  consider  the sparsity constraint on system matrices. The table below presents a comparison of the proposed algorithm and Oymak's method on the simulated dataset described in Section 5.1.
>
>     | Method | Ours | Oymak's method|
>     |:--------:|:--------:|:--------:|
>     | $A$ | $$ \begin{bmatrix}  0 & 0.900\\\\ 0.897& 0  \end{bmatrix}$$| $$ \begin{bmatrix} 0.869 & -0.1535\\\\ -0.151 &-0.891 \end{bmatrix}$$ |
>     | $B$ | $$ \begin{bmatrix}  4.004& 0\\\\0 &4.024  \end{bmatrix}$$| $$ \begin{bmatrix} -1.514 & -1.298\\\\ -1.223& 1.452 \end{bmatrix}$$ |
>     | $C$ | $$ \begin{bmatrix}  1.001 &0\\\\ 0 & 1.011  \end{bmatrix}$$| $$ \begin{bmatrix} -1.289 & 1.432\\\\ -1.525& -1.241 \end{bmatrix}$$ |
>     | $D$ | $$ \begin{bmatrix}  1.473 &0\\\\0 &1.480  \end{bmatrix}$$| $$ \begin{bmatrix}   5.696 & 0.194 \\\\ 0.124& 5.561 \end{bmatrix}$$ |
>     | MRE | **7.35%**|17.22\%|
>
>     The table below presents the mean relative error (MRE) between the predicted outputs of all the learned LDSs and real ones on the considered real-world datasets.
>
>     | Real-world  dataset | Ours | Oymak's method|
>     |:--------:|:--------:|:--------:|
>     |Industrial evaporatoration systems|**13.74\%**|Inf|
>     |Glass furnaces|**18.74\%**|31.21%|
>     | Steam generator |**21.45\%**|76.26\%|
>
>     As observed from the tables, the experimental results on both simulated and real-world datasets demonstrate that the proposed algorithm outperforms Oymak's method. In addition, the system matrix $A$ learned by the Oymak's method on industrial evaporatoration systems is unstable, leading to the corresponding MRE diverging towards infinity.
>
>     **References**
>
>    Samet Oymak and Necmiye Ozay. Non-asymptotic identification of LTI systems from a single trajectory. In 2019 American control conference (ACC), pp. 5655–5661.  2019.

---

### Meta-Review · Area_Chair_iEuE · 2024-12-18

**Metareview:**

This work addresses the challenge of learning sparse system matrices in linear dynamical systems (LDSs), a key tool for modeling time-series data. By imposing sparsity-promoting priors and using the expectation–maximization (EM) algorithm, it estimates both hidden states and system matrices from noisy observations. Additionally, it corrects a common issue in gradient-based methods by ensuring symmetry in noise covariance matrices during optimization.

The paper presents an interesting method for learning LDSs using EM methods. However, the reviewers raised several major concerns:

1. One reviewer noted the absence of meaningful theoretical convergence results for the proposed EM method.
2. Several reviewers criticized the lack of comprehensiveness in the experiments and pointed out that the paper overlooks a rich body of literature on system identification and LDS learning.
3. Another reviewer found the paper's contributions to be incremental.

The AC shares these concerns and believes the paper would benefit from a major revision to address the highlighted issues.

**Additional Comments On Reviewer Discussion:**

The AC engaged in a thorough discussion with the reviewers and concluded that, while the paper has valuable contributions, it requires substantial revision beyond what is feasible within the submission timeline. The AC strongly encourages resubmission, recognizing the paper's importance and potential impact. This was truly a "borderline" decision, and the AC appreciates the authors' efforts and contributions, and hopes they remain encouraged, as the paper has strong merits and can make a significant contribution to the community once revised.

---

### Decision · Program_Chairs · 2025-01-22

Reject